# Environmental co-benefits and adverse side-effects of alternative power sector decarbonization strategies

Gunnar Luderer [1,2]*, Michaja Pehl [2], Anders Arvesen [3], Thomas Gibon [3,4], Benjamin L. Bodirsky [1], Harmen Sytze de Boer[5], Oliver Fricko[6], Mohamad Hejazi[7], Florian Humpenöder [1], Gokul Iyer [7], Silvana Mima[8], Ioanna Mouratiadou [1,9], Robert C. Pietzcker[1], Alexander Popp[1], Maarten van den Berg[5], Detlef van Vuuren [5,9] & Edgar G. Hertwich [3,10]

A rapid and deep decarbonization of power supply worldwide is required to limit global warming to well below 2 °C. Beyond greenhouse gas emissions, the power sector is also responsible for numerous other environmental impacts. Here we combine scenarios from integrated assessment models with a forward-looking life-cycle assessment to explore how alternative technology choices in power sector decarbonization pathways compare in terms of non-climate environmental impacts at the system level. While all decarbonization pathways yield major environmental co-benefits, we find that the scale of co-benefits as well as profiles of adverse side-effects depend strongly on technology choice. Mitigation scenarios focusing on wind and solar power are more effective in reducing human health impacts compared to those with low renewable energy, while inducing a more pronounced shift away from fossil and toward mineral resource depletion. Conversely, non-climate ecosystem damages are highly uncertain but tend to increase, chiefly due to land requirements for bioenergy.

[1] Potsdam Institute for Climate Impact Research (PIK), P.O. Box 60 12 0314412 Potsdam, Germany. [2] Chair of Global Energy Systems, Technische Universität Berlin, Straße des 17. Juni 135, 10623 Berlin, Germany. [3] Industrial Ecology Programme and Department of Energy and Process Engineering, Norwegian University of Science and Technology (NTNU), Høgskoleringen 5, 7034 Trondheim, Norway. [4] Luxembourg Institute of Science and Technology (LIST), 41 rue du Brill, L-4422 Belvaux, Luxembourg. [5] PBL Netherlands Environmental Assessment Agency, Bezuidenhoutseweg 30, The Hague, The Netherlands. [6] International Institute for Applied Systems Analysis (IIASA), Schlossplatz 1, 2361 Laxenburg, Austria. [7] Joint Global Change Research Institute, Pacific Northwest National Laboratory, 5825 University Research Court Suite 3500, College Park, MD 20740, USA. [8] Université Grenoble Alpes, CNRS, INRA, Grenoble INP, GAEL, 38000 Grenoble, France. [9] Copernicus Institute for Sustainable Development, Utrecht University, Princetonlaan 8a, 3584 CB Utrecht, The Netherlands. [10] Center for Industrial Ecology, School of Forestry and Environmental Studies, Yale University, New Haven, CT, USA. *email: luderer@pik-potsdam.de

The international community has agreed to limit global warming to well below 2 °C, and to reach net greenhouse gas (GHG) emissions neutrality in the second half of the twenty-first century[1]. Electricity supply is the single most important emissions source sector, accounting for around 40% of global energy-related $CO_2$ emissions[2]. It also offers the largest low-cost potential for emissions reductions, and thus cost-optimal strategies for keeping global warming to below 2 °C typically feature near-zero electricity sector emissions by mid-century, and rely increasingly on electrification to minimize fossil fuel use in the transport, industry and buildings sectors[3–6].

Beyond economic costs and GHG emissions, sound climate policies also have to take into account other sustainability dimensions, such as those laid out in the UN's Sustainable Development Goals (SDGs) adopted by the United Nations in 2015[7]. The energy sector is the origin of a wide variety of environmental impacts. While much of the public debate focuses on its contribution to global warming via greenhouse gas emissions, energy supply systems also account for substantial shares of other environmental impacts, such as air and water pollution[8,9], land occupation[10], water use[11–14], ionizing radiation and nuclear waste[15], as well as fossil and mineral resource depletion[16]. Energy system futures therefore are particularly relevant for SDGs 3 (health), 6 (clean water), 12 (responsible consumption and production), 14 (life below water) and 15 (life on land).

Thus far, there is only very limited system-level research on the benefits and adverse side-effects of future decarbonized power supply in terms of nonclimate environmental impacts. Process-detailed integrated assessment models (IAMs) of the energy-economy-climate system are frequently used to analyze alternative climate change mitigation strategies and their implications, with a focus on greenhouse gas emission reductions. Only recently other specific environmental impacts such as air pollution[8,17], land-use for bioenergy[18,19] or water demand[11,12,20,21] have been included in IAMs, but so far none of these studies considers the breadth of impacts studied here. Accordingly, a consistent and holistic evaluation of co-benefits of different mitigation pathways is still missing.

By contrast, life-cycle assessment (LCA) as conducted by the industrial ecology community tracks a variety of substance flows[22] and considers a broad set of environmental impacts[23]. But most LCA focuses on current technology and production systems, not accounting for changes in environmental performance of individual technologies (e.g., due to technological learning in the case of photovoltaics), or due to large-scale and structural systemic changes (e.g., a switch towards low-carbon technologies in the context of 2 °C climate stabilization). Some progress towards *prospective LCA* incorporating future technological changes has been made, using various techniques and approaches for the power sector[9,24–26], and increasingly also end-use technologies[27,28]. Several studies have performed an ex-post LCA of future national or regional energy scenarios[29,30], or applied an ex-post LCA to global International Energy Agency scenarios[9,31]. Other studies have investigated the importance of indirect life-cycle greenhouse gas emissions for optimal decarbonization pathways[25,32,33], but do not consider other environmental impacts.

The novel contribution and main goal of our study is to combine global IAM and LCA approaches, drawing on their specific strengths, to quantify a wide variety of environmental co-benefits and adverse side-effects of a portfolio of alternative power sector decarbonization pathways. In contrast to earlier studies[9,29] we also comprehensively cover all major power technologies, while fully and consistently accounting for system-level interdependencies, future technological change and decarbonization of the supply chain in the context of a global 2 °C climate stabilization effort.

We find that the transition to low-carbon power systems has major co-benefits across a large number of environmental impacts, in particular those related to human health, ecotoxicity and fossil resource use. Land requirements and mineral resource depletion are exacerbated in decarbonization pathways and thus emerge as crucial sustainability trade-offs with climate change mitigation. Our study shows that the scale and profile of co-benefits and adverse side effects depend strongly on technology choice. Co-benefits tend to be greatest for decarbonization strategies focusing on renewable energy technologies.

## Results

**Alternative power system decarbonization pathways.** There are a number of viable technology options for low-carbon electricity supply. Consequently, there is much more flexibility in decarbonizing the power sector than nonelectric energy supply[5]. For instance, some low-carbon electricity pathways rely heavily on nuclear or carbon capture and storage (CCS), while others focus mostly on renewable energy sources[5,34]. This is reflected in the scenario set considered here: In addition to a Full Technology climate change mitigation scenario (FullTech), we consider two more mitigation scenarios with either a Conventional Technology portfolio (combined share of wind and solar power restricted to 10%—Conv, in line with assumptions in other IAM studies that explored techno-economic impacts of technology constraints[6,35,36]), or a New Renewables portfolio (nuclear phase-out, no CCS deployment in the power sector—NewRE) to contrast the implications of opposing mitigation strategies. Moreover, considering a baseline scenario Base without emissions constraint establishes a reference point against which we can evaluate co-benefits and adverse side-effects of power sector decarbonization (Table 1). The resulting electricity generation mixes for the four scenarios and five participating IAMs are shown in Fig. 1 and available in Supplementary Data 1.

**Impacts on human health.** The energy sector poses health risks due to emissions of air, water, and soil pollutants, in addition to those of greenhouse gases. Our analysis framework allows us to contrast these impacts for the four transformation scenarios and to attribute them to specific technologies. It should be noted that the life-cycle assessment approach limits our evaluation to normal operation only, which means that impacts from exceptional events

---

**Table 1 Overview of scenarios considered**

| Name | Short | Carbon constraint | Technology availability |
|---|---|---|---|
| Baseline | Base | No emissions constraint | Full portfolio |
| Full technology portfolio | FullTech | Cumulative 2011−2050 power sector emissions limited to 240 | Full portfolio |
| Conventional technology | Conv | GtCO₂. | Wind and solar power limited to 10% |
| New renewables | NewRE | | Nuclear phase-out, no CCS in the power sector |

We consider three mitigation scenarios consistent with the 2 °C warming limit with different power sector technology portfolios. In addition, a Baseline without emissions constraint serves as a reference point against which co-benefits and adverse side effects of climate policies can be evaluated

---

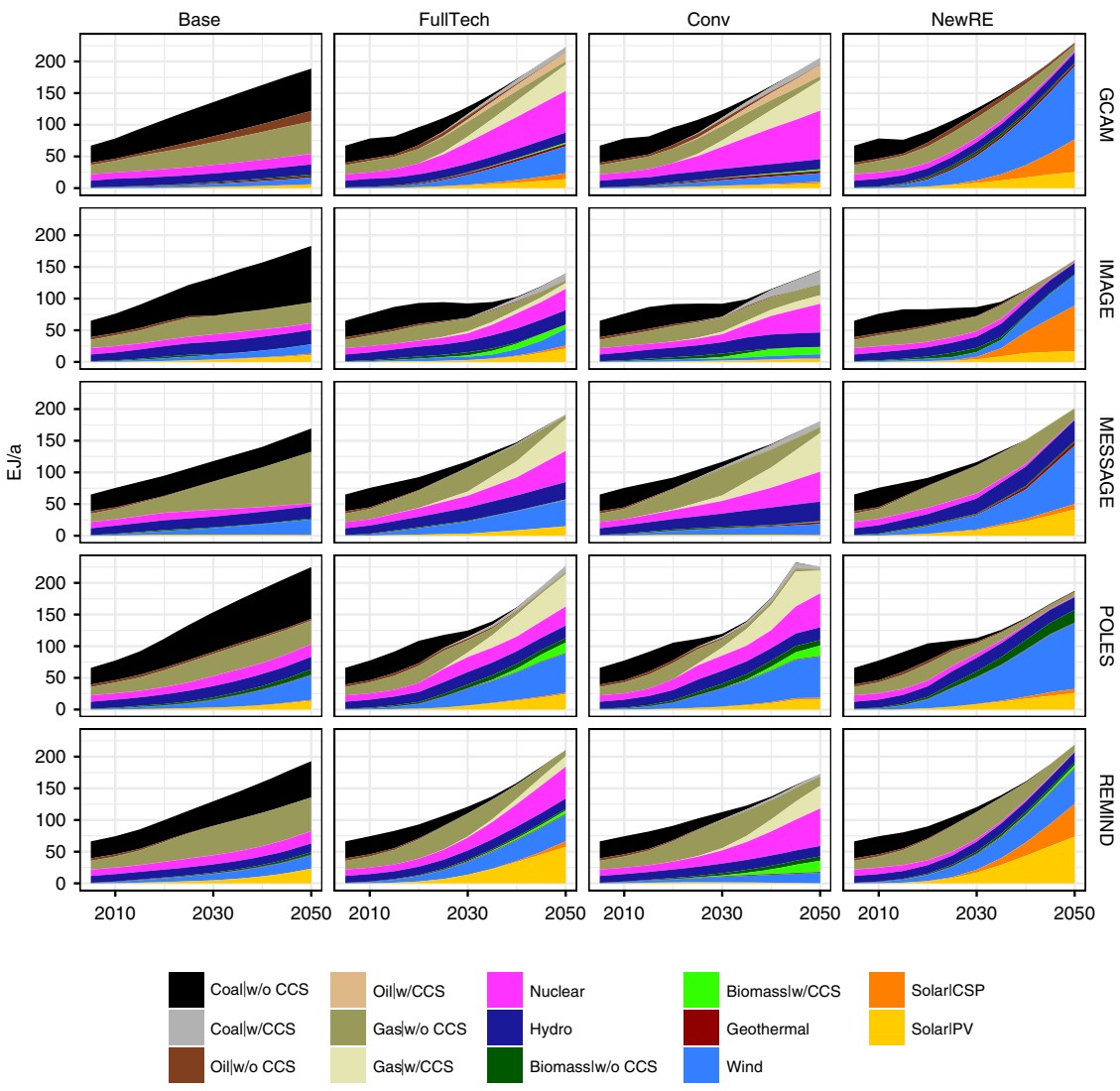

**Fig. 1** Scenarios of future electricity generation. Projections of electricity generation for the baseline and three decarbonization scenarios compatible with limiting warming to well below 2 °C, as modeled by five structurally different Integrated Assessment Modeling systems

(e.g. dam failures, mining accidents, pipeline explosions, or nuclear meltdowns) are excluded from this analysis.

The power sector is one of the major sources of air pollutant emissions, and air pollution is a major threat to human health[37]. In 2010, the power sector accounted for around 40% for global $SO_2$ emissions, and 20% of $NO_x$[38]. These substances are important precursors for particulate matter formation (PM-10) (Fig. 2a). $NO_x$, along with $CH_4$ and other volatile organic compounds (NMVOCs), also enhance photochemical oxidant formation, i.e., tropospheric ozone (Fig. 2b). In particular PM-10 but also tropospheric ozone are important health threats[39]. In line with previous studies[8,17], we here find that air pollutant emissions and concentrations stabilize or decrease slightly even with a massive upscaling of fossil-based power production in absence of climate policies (Fig. 2a). This is largely due to increasing regulation and end-of-pipe measures to control pollution[8,40] and follows the historical trend in industrialized countries[38]. The power sector's contribution to PM-10 and ozone originates almost exclusively from the combustion of fossil fuels and bioenergy (Fig. 2a, b). Our analysis also accounts for upstream emissions due to indirect energy demands for the construction of energy conversion technologies, fuel production and handling. However, we find that upstream fossil fuel use[25] and indirect air pollution associated with noncombustion power

technologies are rather small compared to direct emissions (see Supplementary Fig. 1).

Climate change mitigation lowers air pollution drastically and more so in the NewRE than in Conv scenarios. On average across models, the decline of fossil-based power in NewRE climate policy results in reductions of 87% and 83% of PM-10 and ozone precursors, respectively, relative to the baseline case. Air pollution impacts in the Conv case are around double of those in the NewRE case, largely due to greater remaining direct $NO_x$ emissions as well as higher indirect air pollution from upstream energy requirements for the extraction and handling of fossil fuels. This is despite strong co-control of sulfur in CCS plants (Fig. 3, Supplementary Fig. 2 and ref. [41]).

All energy technologies cause human toxicity impacts due to chemical toxicant emissions in their supply chains, albeit at different scales (Fig. 2c). They are particularly high for coal (leaching of toxicants from mine dumps), bioenergy (agrochemicals use in agriculture), and still significant for gas (emissions during natural gas extraction), nuclear (tailings from uranium mining and milling) and photovoltaics (emissions from copper processing and silicon refinement). Overall, a pattern similar to air pollution impacts emerges: human toxicity is strongly reduced under climate policies, and around 60% lower in NewRE compared to Conv.

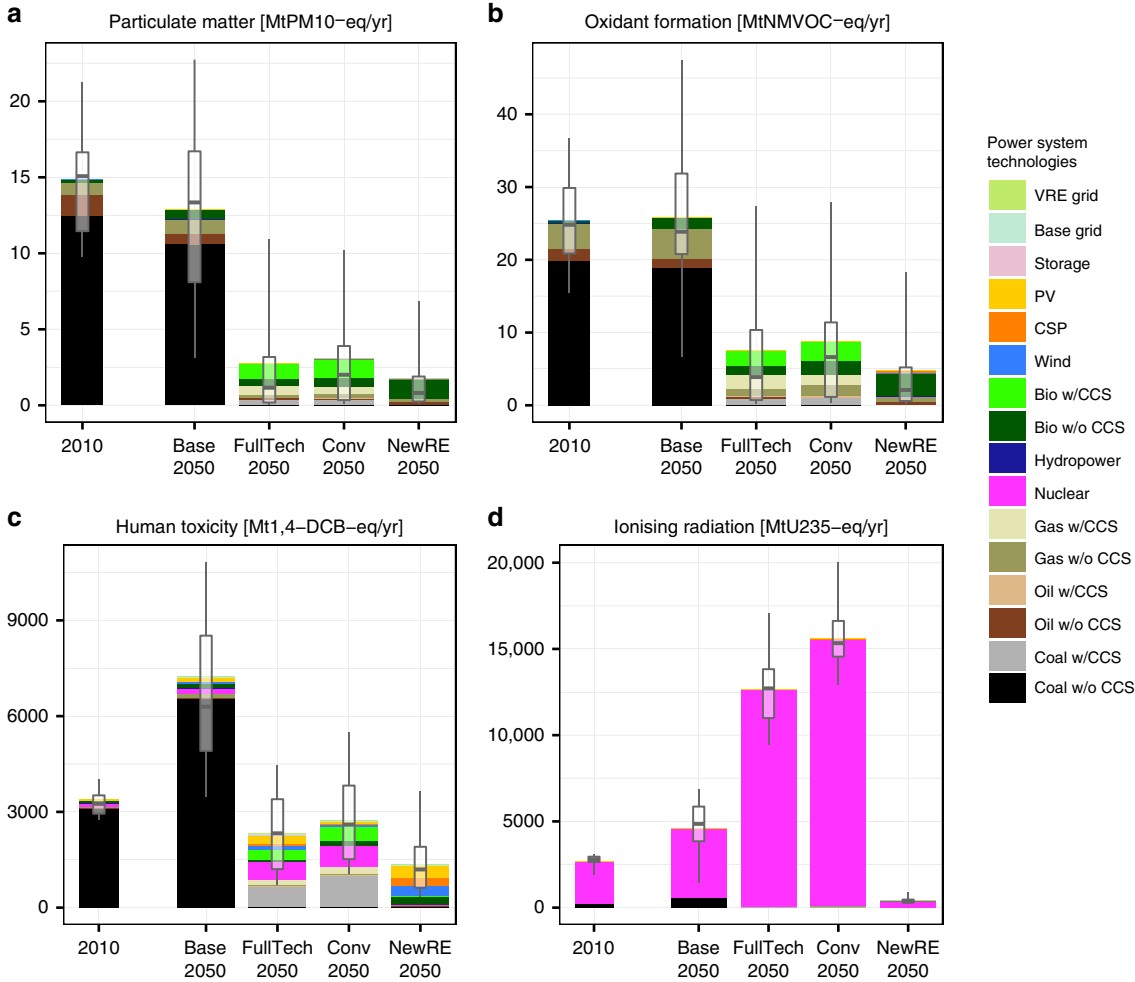

**Fig. 2** Environmental impacts affecting human health. Globally aggregate environmental impacts from **a** particulate matter formation, **b** photochemical oxidant formation, **c** human toxicity, and **d** ionizing radiation, in 2010 and 2050 under different power sector transformation scenarios. Stacked bars indicate mean across all combinations for LCA technology variants and IAM scenario realizations. Boxplots indicate median and interquartile ranges across technology variants and participating integrated assessment models, whiskers 10th−90th percentile ranges. Ranges do not reflect uncertainty in environmental impact characterization. Base Grid refers to generic grid requirements determined by total electricity demand, while VRE grid refers to additional grid requirements for coping with the variability of renewable electricity supply from wind and solar power

Another relevant impact from the power sector stems from the ionizing radiation emitted by radioactive substances (Fig. 2d). Ionizing radiation is almost exclusively caused by nuclear power, and dominated by releases from mining and milling during the production of nuclear fuels. Per-unit impacts for all other technologies, including coal power, are more than two orders of magnitude smaller (see Supplementary Fig. 2) and largely due to upstream nuclear power use. Importantly, LCA inventories and assessment methods do not account for the risk of radiation exposure nuclear accidents[42]. However, analysis by Hirschberg et al.[43] and others[44,45] suggest first that, in terms of lost life years, fatalities from accidents tend to be considerably smaller than health impacts from regular operation, and second that fatalities from nuclear accidents tend to be lower than those from fossil-based or hydropower.

In the absence of climate policies, models project an increase of around 50% nuclear power use by 2050, resulting in a corresponding rise in related radiation impacts. Climate change mitigation could result in a further expansion of nuclear power and corresponding radiation impacts by a factor of 3–7 in the FullTech scenarios relative to 2010, or even 5–8 if the use of wind and solar power is limited (Conv scenarios). In the NewRE scenarios, by contrast, ionizing radiation impacts are limited to

the extent that pre-existing nuclear power plants are phased out of power supply. The analysis of endpoint impacts indicates that ionizing radiation contributes less to human health impacts than particulate matter, ozone, or other toxic pollution (combined assessment section).

**Ecosystem damage**. The power sector also threatens the health of ecosystems. Relevant impact channels include land occupation and transformation, as well as pollutant release resulting in terrestrial acidification, eutrophication and ecotoxicity impacts.

Land-use for agricultural and other human activities is a crucial driver of global biodiversity loss and degradation of many ecosystem services[46]. In 2010, the land footprint attributable to power supply compared to around 12% of total built-up area[47]. The ReCiPe LCIA differentiates between land occupation of areas already transformed from its natural state, and natural land transformation, e.g. from forests to croplands. Natural land transformation accounts for the quality of the land being transformed, putting particular emphasis on reduction of biodiversity-rich forest areas. In all scenarios considered, both land occupation (Fig. 4a) and natural land transformation (Fig. 4b) for power supply will increase in the future relative to

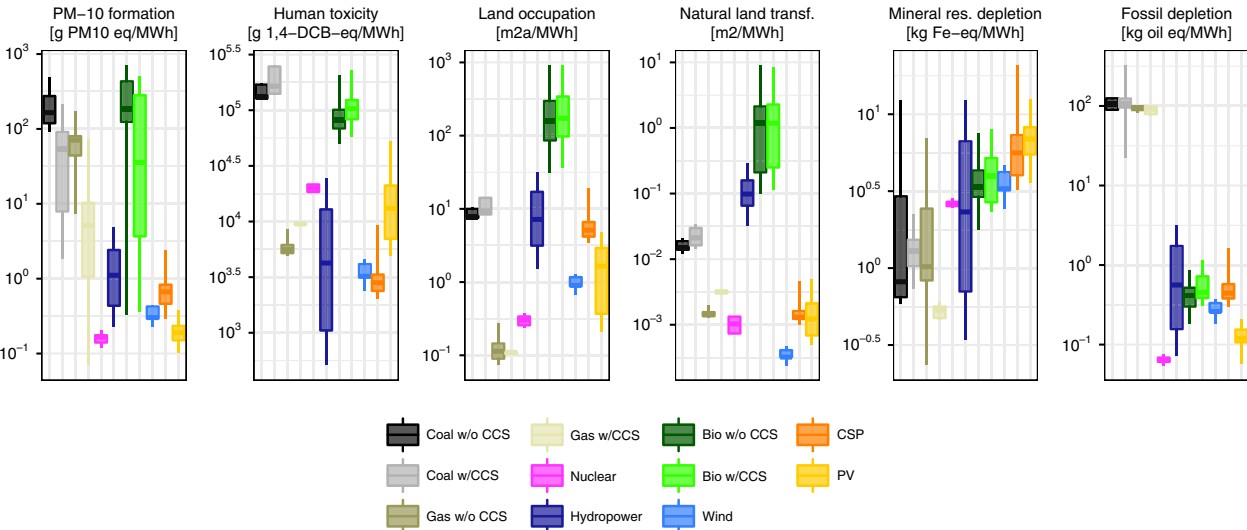

**Fig. 3** Selected technology-specific environmental impacts. Per unit life-cycle impacts of electricity technologies for the FullTech scenario and the year 2050 for impact indicators dominating the endpoints human health, ecosystem damages and resource depletion. Boxplots indicate median and interquartile ranges across technology variants and participating integrated assessment models, whiskers 10th−90th percentile ranges. Ranges do not reflect uncertainty in environmental impact characterization. The full set of indicators is displayed in Supplementary Fig. 2

current level. In Base, the power sector's land-use increases due to an increase of the power system's scale and is largely attributable to coal (both area occupied by open-cast coal mines, and land-use associated with timber used for the support of underground mines), biomass and hydropower (land-use for reservoirs). We find that climate policy tends to increase power-system related pressure on land, largely because of increasing biomass use. On a per-MWh basis, electricity from biomass with CCS is more than 20 times more land-intensive than hydropower, coal with CCS, or CSP, and exceeds wind and PV by around two orders of magnitude (Fig. 3). Due to the deforestation induced by biomass expansion, bioenergy figures even more prominently in natural land transformation impacts. Overall, bioenergy-induced land-use impacts tend to be greatest in the Conv scenarios, as negative emissions from BECCS are required to compensate for residual $CO_2$ from imperfect carbon capture in fossil CCS plants. Importantly, however, we find very high uncertainty—i.e., technology, policy and management dependence—in the ecosystem impacts form land-use, with the variability induced by management practices and IAM model uncertainty exceeding the differences across scenarios (Supplementary Fig. 3). In comparing fossil to nonfossil power generation, it is also important to emphasize that our analysis does not account for habitat losses caused by coastal flooding. Due to higher climate change-induced sea level rise, coastal flooding will be more severe in the Base scenario than in the climate change mitigation scenarios.

Another important factor for the land footprint of electricity supply systems is the grid infrastructure for transmission and distribution, which accounts for around one third of the total. As an expansion of grid interconnectors is an important option for coping with the variability of wind and solar power supply[48], we here account explicitly for the dependence of transmissions grid requirements on the generation share of wind and solar (see Methods). However, we find that the land occupation attributable to additional grid requirements for wind and solar integration is small compared to the land footprint from the general electricity grid.

Further ecosystem damage is inflicted from the release of various chemical substances. Atmospheric sulfur and nitrogen oxides from combustion result in terrestrial acidification. In line with the reduction of health impacts from air pollution, terrestrial acidification is projected to decline slightly under the baseline

scenario, and to fall to less than a fifth of current levels by 2050 under 2 °C-consistent stabilization (Fig. 4c). Similar to toxicants harmful to humans, all technologies feature life-cycle ecotoxicity impacts (Fig. 4d). However, on a per-MWh basis, these are greatest for fossil technologies (emissions during extraction), substantial for bioenergy (agrochemicals use for crops), and much smaller for wind and solar (Supplementary Fig. 2). As a consequence, ecotoxicity impacts in the NewRE decarbonization scenarios are around 30% lower than those in FullTech. As the Conv scenarios rely more heavily on natural gas with CCS, ecotoxicity impacts are 25% greater than in FullTech, and on average around double those estimated for 2010.

Another relevant channel for ecosystem impacts are marine and freshwater eutrophication (Fig. 4e, f). The leaching of phosphate from coal production is the dominant contributor to freshwater eutrophication impacts, as ReCiPe assumes phosphate to be the primary limiting nutrient for freshwater ecosystems[23]. In contrast to freshwater, nitrates induce a higher eutrophication response for marine ecosystems[23,49]. Emissions of nitrogen oxides from combustions as well as direct nitrate releases from fertilizers for bioenergy cultivation therefore contribute to marine eutrophication. For both freshwater and marine eutrophication, the strong reduction of fossil fuel use results in substantial decreases in mitigation scenarios compared to Base. These co-benefits are greatest for the NewRE scenarios.

Not only the contamination of water with chemical substances, but also its withdrawal from river systems is an important environmental stressor. Electricity supply systems account for approximately 14% of global human water withdrawal[13]: Most thermal power plants use water for cooling, while hydroelectric plants affect waterways through dams and water losses to evaporation and seeping[50,51]. As discussed in earlier literature[11,12,20,50,51], future projections of water withdrawals are highly uncertain as they depend on the degree to which utilities adapt to water scarcity, for instance by installing dry cooling technologies in thermal power plants. Besides cooling water, water losses from hydropower and withdrawals for biomass irrigation are projected to increase substantially in the future[21]. Across decarbonization scenarios, water withdrawal is highest in the Conv scenarios due to the large share of nuclear power, which is particularly cooling-water-intensive (Fig. 4g). The NewRE scenarios, by contrast,

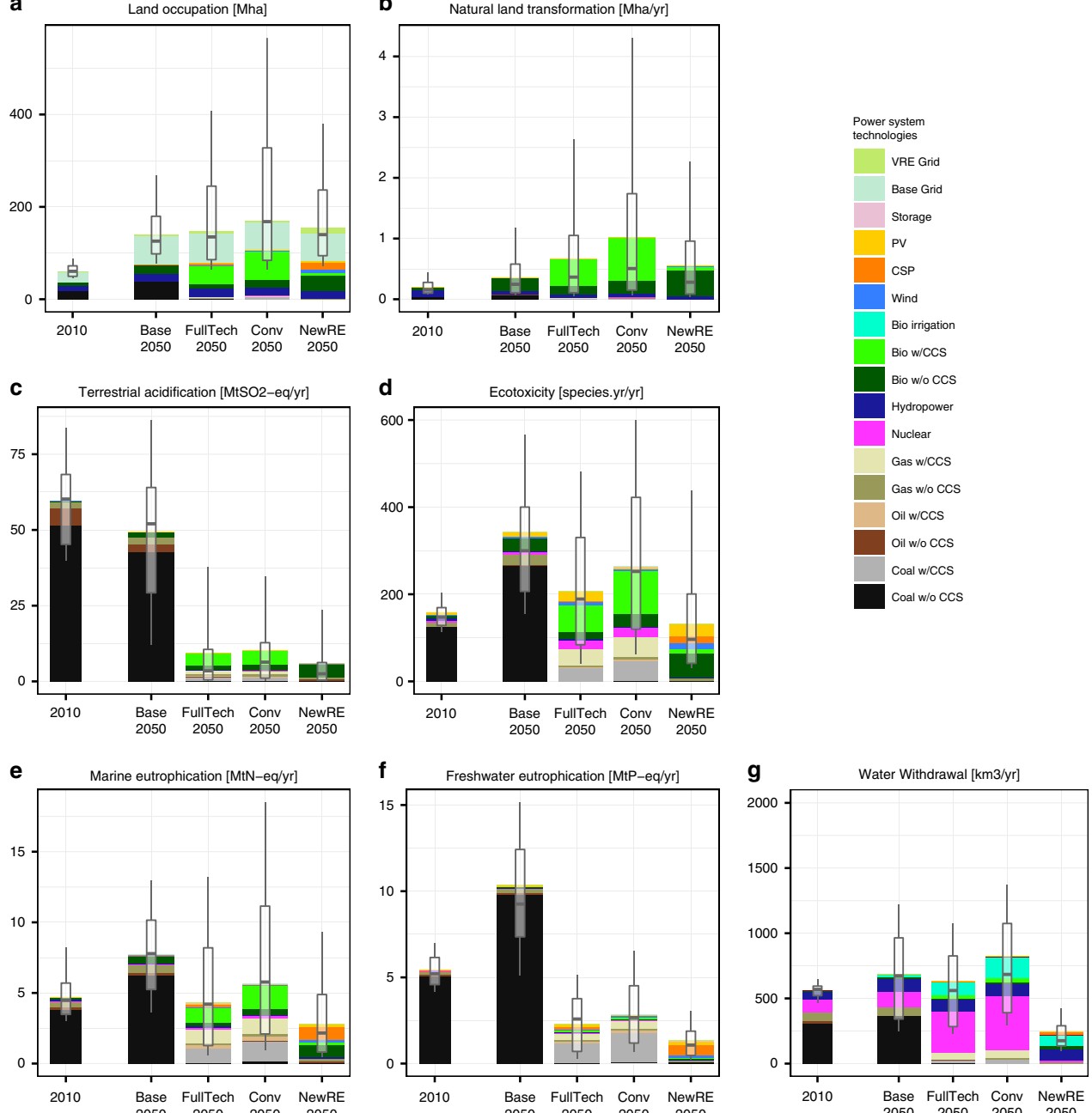

**Fig. 4** Environmental impacts affecting ecosystems. Globally aggregate environmental impacts from **a** land occupation, **b** natural land transformation, **c** terrestrial acidification, **d** freshwater ecotoxicity, as well as **e** marine and **f** freshwater eutrophication, as well as **g** water withdrawal, for 2010 and 2050 under different power sector transformation scenarios. Ecotoxicity was calculated as the aggregate of terrestrial, marine and freshwater ecotoxicity in units of species-years using characterization factors from ref. [23]. Stacked bars indicate mean across all combinations for LCA studies and IAM scenario realizations. Boxplots indicate median and interquartile ranges across technology variants and participating integrated assessment models, whiskers 10th−90th percentile ranges. Ranges do not reflect uncertainty in environmental impact characterization

have very little thermoelectric capacities, and thus features distinctly lower water withdrawals than the Conv and FullTech scenarios.

**Exhaustible geological resources.** Beyond damages to human health and ecosystems, the energy sector also contributes strongly to the depletion of exhaustible resources, thus reducing natural capital and options for future generations. It is important to keep in mind that the health and ecosystem damage associated with resource extraction are already accounted for in the other impact indicators, such as ecotoxicity or human toxicity.

In absence of climate policies, fossil depletion is projected to roughly double by mid-century relative to 2010 levels, as supply-side efficiency improvements and the contributions of renewables and nuclear are insufficient to offset strongly increasing electricity demand (Fig. 5a). Climate policy does not necessarily reduce fossil depletion, as gas with CCS becomes increasingly important and replaces coal. In the Conv and FullTech cases, around half the models project 2050 fossil use for power supply to exceed 2010 levels. In NewRE case, by contrast, the models project on average an around 75% reduction of fossil depletion relative to 2010 levels. Natural gas used to provide flexible backup power compensating

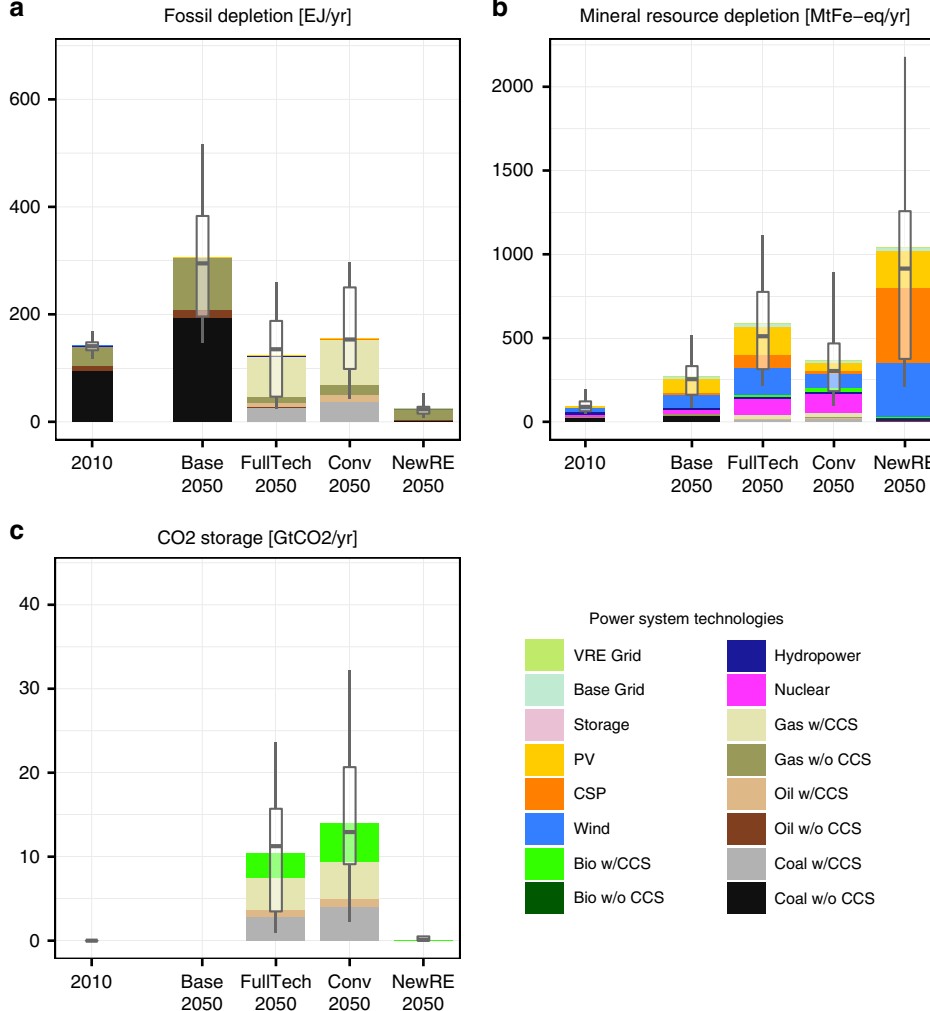

**Fig. 5 Resource impacts.** Globally aggregate resource impacts from **a** fossil depletion, **b** mineral resource depletion, and **c** geological $CO_2$ storage requirements, for 2010 and 2050 under different power sector transformation scenarios. Stacked bars indicate mean across all combinations for LCA studies and IAM scenario realizations. Boxplots indicate median and interquartile ranges across technology variants and participating integrated assessment models, whiskers 10th−90th percentile ranges. Ranges do not reflect uncertainty in environmental impact characterization

fluctuations from wind and solar electricity accounts for most of the remaining fossils in these NewRE scenarios. Indirect fossil energy requirements for power supply, e.g., manufacturing of solar panels, are fully accounted for in our analysis but found to be relatively small even in the NewRE scenario.

In the FullTech and Conv scenarios, the continued use of fossils can only be reconciled with the tight emissions constraints via carbon capture and storage (CCS). This gives rise to geological $CO_2$ storage as a new exhaustible resource depleted by the energy sector. Our results indicate that the power sector would account for around 11 [4−16] $GtCO_2$/yr storage requirements in FullTech, and 15 [10−21] $GtCO_2$ in the Conv scenarios (Fig. 5c) by 2050, which increases further in the majority of model simulations thereafter (Supplementary Fig. 4). The power sector competes with other potentially important CCS use cases, such as biomass with CCS for nonelectric fuels, CCS for industry, or direct air capture. While currently available estimates suggest a total geological technical potential for $CO_2$ storage of at least ~2000 $GtCO_2$[52], economically and societally acceptable $CO_2$ storage potentials are likely to be much more limited.

Power supply also accounts for a substantial share of mineral resource depletion, mostly for the construction of power generators. In 2010, around 5% of global copper, 2.5% of aluminum, and 3% of iron went into the electricity supply sector[53]. Mineral resource depletion accounts for the aggregate demands from these bulk metal demands along with some 20 other important mineral resources. It should be noted that concerns about mineral resource depletion involve a large number of minerals, not all of which are covered by life-cycle impact assessment methods. For example, the indicator used here does not include neodymium or dysprosium (used in certain wind turbines[54]), or indium or tellurium (used in certain photovoltaic cells)[54]. In all scenarios, nonfuel mineral depletion increases relative to current levels. In contrast to all other indicators we find that all climate policy scenarios feature higher mineral resource requirements, and that in the NewRE scenarios 2050 mineral resource depletion is around twice as high as in FullTech, and around four times higher than in the baseline (Fig. 5b). This is explained, first, by the higher per-unit metal requirements for renewable technologies, particularly solar PV; second, the fact that wind and solar technologies require substantial material upfront investments before operation (which here are attributed to the year of construction); and finally, to a lesser extent, the additional metal resources required for the build-up of additional grid and storage infrastructure to accommodate the variability of wind and solar power supply.

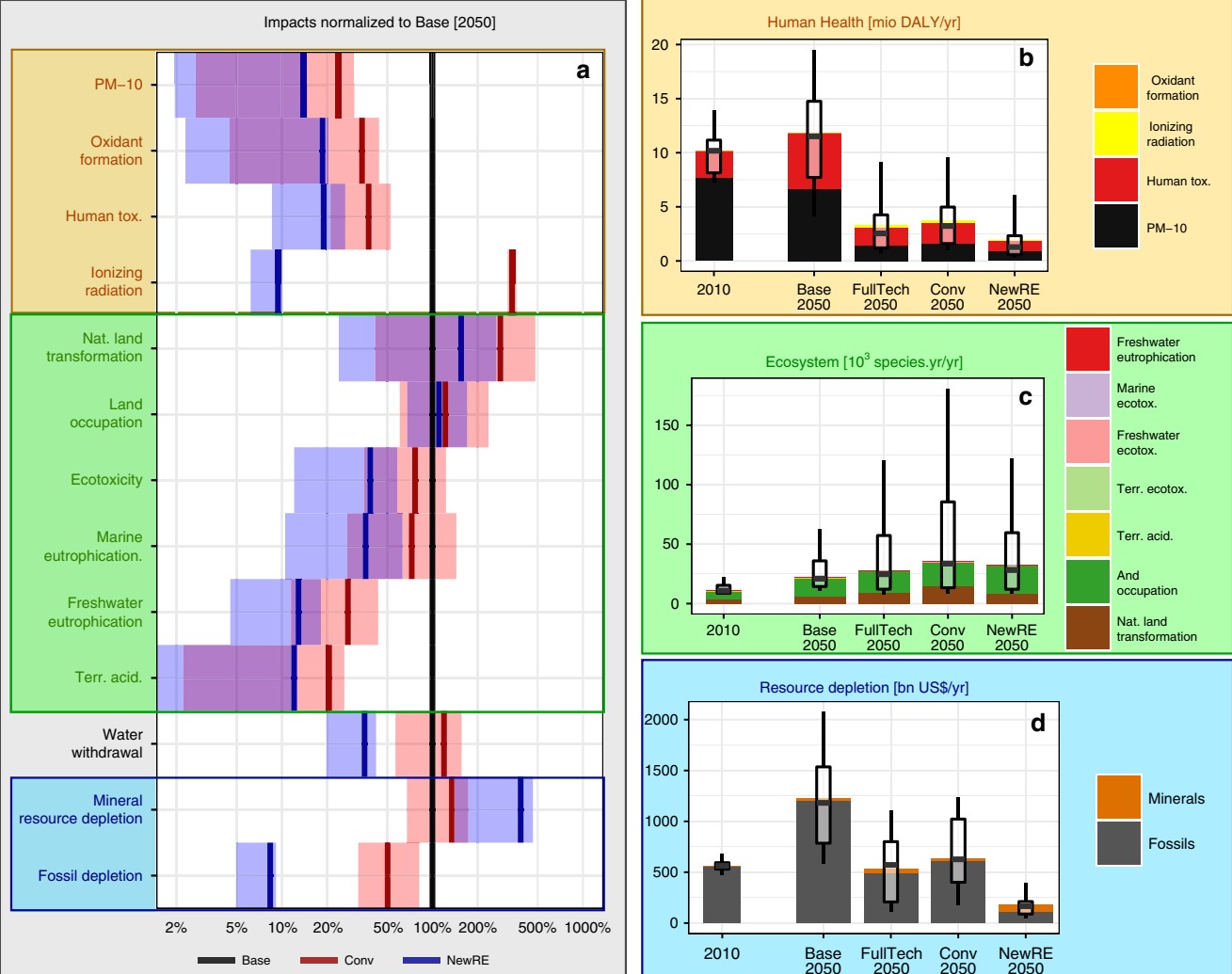

**Fig. 6** Combined assessment of global environmental impacts of alternative decarbonization strategies in 2050. **a** Relative size of midpoint environmental impacts for conventional (Conv scenarios) vs. new renewable-based (NewRE scenarios) strategies, compared to those that would have occurred in absence of climate policies (Base scenario), on a logarithmic scale. Aggregate endpoint impacts by (midpoint) impact channel: **b** human health damages in disability-adjusted life-years lost, **c** ecosystem damage in species-years, and **d** surplus costs from exhaustible resource depletion. Shaded ranges in (**a**) as well as boxplots in (**b−d**) indicate interquartile ranges across IAMs and LCA technology variants, whiskers in (**a−d**) indicate 10th−90th percentile range. Ranges do not reflect uncertainty in environmental impact characterization. Note that oxidant formation, freshwater eutrophication, ecotoxicity and terrestrial acidification impacts account for a below 2% share in total endpoint impacts, and are therefore barely visible

**Combined assessment**. The comparison of differences across scenarios demonstrates that electricity decarbonization has substantial nonclimate co-benefits for most environmental impacts at the midpoint level of the cause-effect chain (Fig. 6a), as well as the human health and resource depletion impacts at the endpoint level (Fig. 6b–d). However, some environmental pressures induced by power supply emerge as crucial concerns, as they are likely to increase in the future and might be exacerbated by the low-carbon transformation: first, land requirements; second, mineral resource depletion; and third, impacts related to the use of radioactive materials, not only ionizing radiation as considered here, but also the risk of nuclear accidents and the production of nuclear waste.

We further find that different decarbonization strategies result in distinctly different profiles of risks and co-benefits. Wind and solar-based decarbonization (NewRE scenario) consistently achieves highest reductions in health-related environmental impacts (Fig. 6b). Fossil technologies—especially coal—dominate aggregate health impacts by far (see Supplementary Fig. 5); thus,

their faster and deeper phase-out in the NewRE scenarios yields greatest benefits, with around 60% lower aggregate mortality compared to Conv, and an around 50% decrease relative to FullTech in 2050. The most prominent contributors to health impacts are air pollution and human toxicity.

NewRE decarbonization also minimizes pollution-related ecosystem impacts compared to Conv and FullTech scenarios. Aggregate ecosystem damage, as derived from the corresponding ReCiPe endpoint characterization factors[23], are dominated by land occupation and natural land transformation. These land-use related impacts are highly uncertain and of comparable magnitude across the different decarbonization scenarios: While NewRE scenarios are characterized by greater land-requirements for wind and solar power as well as grid expansion, the higher bioenergy deployment in the Conv scenarios induces greater natural land transformation (Fig. 6c and Supplementary Fig. 5).

We also find that decarbonization will fundamentally change the resource requirements of the power sector, away from fossil fuel inputs and towards mineral resources (FullTech and NewRE)

and geological storage space for $CO_2$ (FullTech and Conv). For the NewRE scenarios in 2050, fossil depletion decreases by 90%, while bulk material requirements increase four-fold compared to baseline levels. In addition, certain wind power and photovoltaics technologies also rely on specialty minerals, such as dysprosium or indium[55,56], which are not addressed in the resource depletion assessment method employed here, but are subject to geopolitical supply risks[57]. The low-carbon transformation, especially if it relies heavily on wind and solar technologies, can be expected to have profound implications for the geopolitical landscape, pointing to the need for flanking the global clean energy effort with an integrated critical materials strategy.

Fossil fuels by far dominate resource surplus costs, the aggregate ReCiPe endpoint indicator for resource depletion (Fig. 6d). This result suggests that the benefit to society stemming from reduced fossil requirements in NewRE outweigh the burden due to additional mineral resource depletion. In addition, it should be kept in mind that much of the 2050 resource requirements for wind and solar installations can be attributed to upfront investment for electricity produced later, and that mineral resources are amenable to recycling[58], while fossil resources are not.

In terms of technologies, fossil fuels are the major drivers of health impacts and also dominate resource surplus costs; thus, their reduction in the context of climate policies yields substantial benefits (Supplementary Fig. 5). Bioenergy emerges as the greatest driver of ecosystem damage, chiefly due to land occupation and induced loss of natural lands. On the other hand, numerous studies have demonstrated the importance of bioenergy for the 1.5 and 2 °C targets[3–5,18,59], both due to its versatility in substituting fossil fuels and the possibility of generating negative emissions. This underlines the need for an integrated global land management to navigate the tradeoff between climate change mitigation and conservation.

## Discussion

The world is currently witnessing a dynamic and robust growth of wind and solar power, which is also expected to become the most important contributor towards near-term $CO_2$ reduction efforts worldwide[60]. Our results suggest that further relying predominantly on these new renewables in the transition towards a near-zero emissions power system also reduces most nonclimate environmental impacts on the system level compared to strategies that limit the contribution of wind and solar power largely in favor of greater CCS deployment.

It is important to bear in mind that our forward-looking global analysis with wide system boundaries, despite the methodological advancements brought by integrating integrated assessment models and prospective life-cycle assessments, is subject to significant limitations and uncertainties. For example, the linearized approach of life-cycle impact assessment cannot account for scale-dependent variations in per-unit impacts, e.g., due to threshold or saturation effects, or interaction among different environmental impacts. Human toxicity and ecosystem impacts are subject to spatial variability. Changes in population and age structure matter for health damages, ecosystem damage will depend on future land-use patterns, and the economic consequences of resource depletion on competing resource uses. Our study accounts for dynamic changes in technical systems (e.g., increased material efficiency of PV cells, or reduction of air pollution due to end-of-pipe measures), but lacks a dynamic description of crucial nonclimate environmental mechanisms, mostly due to a lack of knowledge or demonstrated importance of relevant developments. While our analysis accounts for uncertainties in energy technology deployment as well as innovation in individual technologies, we were not able to account for uncertainties in the characterization factors translating stressor flows to environmental impacts (see Methods).

We deliberately focused our analysis on the year 2050, since by mid-century the decarbonization of power systems is largely completed and technology developments get increasingly uncertain with longer time horizons. Nonetheless, it is important to note that environmental impacts of a decarbonized power system might continue to evolve thereafter, depending on size and composition of power supply. For instance, increasing contributions of biomass to electricity generation, as projected in many IAM scenarios[61], will exacerbate ecosystem damages.

LCA impact assessments develop rapidly. For instance, a recently published update of ReCiPe[62] includes spatially explicit characterization factors. We here presented an approach for evaluating power sector transformation pathways in terms of a wide spectrum of environmental impacts. However, it thus far does not consider regional variations in impact assessment parameters. It also does not account for the effect of nonclimate environmental impacts on optimal policy choice. This would require formulating limits on each of the impacts under consideration, or monetary valuation of associated damages. Building on the LCA and IAM coupling method presented here, future research will further integrate emerging advances in life-cycle assessment, with energy-economic modeling as well as land-use and biodiversity models to derive increasingly robust science based decision support for a sustainable low-carbon energy transformation.

## Methods

**Overview of modeling approach**. Our study incorporates a number of innovative methodological features. By using five structurally different IAMs (GCAM, IMAGE, MESSAGE-GLOBIOM, POLES, REMIND) we are able to capture diversity in system transformations for given climate and technology policy assumptions. The IAMs represent several environmental impact mechanisms directly and by source, in particular air pollution (resulting in particulate matter formation, oxidant formation and terrestrial acidification), water use for cooling and hydropower as well as fossil resource depletion. Land requirements, fertilizer use, irrigation water and land-use change emissions from bioenergy production have been estimated from the MAgPIE land-use management model[63], allowing to account for indirect effects of bioenergy (such as induced deforestation) in a dynamic setting. In addition, we estimate requirements for power transmission grids and electricity storage using a regression from the detailed power system models REMIX and DIMES[48,64]. We combine IAM-based technology deployment pathways with per-unit environmental impact and indirect energy requirements derived from the multiregional LCA model THEMIS[24,26]. This allows accounting for future technological change, e.g. in terms of increased efficiency, changes of energy supply mixes, biomass supply systems, or material demands for solar PV (see Methods). We further use the ReCiPe life-cycle impact assessment (LCIA) methodology[23], which allows us on the one hand to expand the assessment to include human toxicity, ionizing radiation, land occupation, terrestrial, freshwater and marine ecotoxicity, marine and freshwater eutrophication, as well as mineral resource depletion, and on the other hand to not only cover direct, but also indirect environmental impacts associated with manufacturing or other supply chain activities. In an additional step, in line with the established ReCiPe LCIA[23], we aggregate the above-mentioned impacts (midpoints along the cause-effect-chain) to the three endpoint categories human health, ecosystem damage and resource depletion. Importantly, the endpoint impacts are subject to much greater uncertainty than midpoints. We therefore use endpoint results mainly as an indicative guide to the relative severity of the various impacts, and for relative comparison of different decarbonization pathways. Importantly, we note that the ranges provided in the figures and quantitative results only reflect uncertainties in technology choice and technological development of individual technologies, but not the uncertainty from environmental impact characterization, which is unavailable in the ReCiPe methodology. A detailed description of the approach is presented in Methods.

**IAM scenarios and technology deployments**. To characterize alternative decarbonization strategies in terms of their environmental impact, we combine IAM scenarios with LCA data for specific power sector technologies. To account for the uncertainties about future developments, these scenarios were run by the five integrated assessment models GCAM, IMAGE, MESSAGE-GLOBIOM, POLES and REMIND. These five IAMs are well-established modeling systems and have participated in numerous prior multi-model studies of long-term and global energy transformation pathways (e.g., ref. [3]). They are characterized by a broad coverage

of power sector technologies and process detail in the energy system. All five models represent air pollution and water requirements by source. A more detailed description of the individual models can be found in the Supplementary Text.

The scenario set comprises one baseline scenario without any climate policies and four alternative climate policy scenarios with different policy choice (Table 1). All climate policy scenarios limit the cumulated 2011−2050 $CO_2$ emissions from the power sector to 240 $GtCO_2$, resulting in an at least 80% reduction of emissions by 2050 relative to 2010, and consistent with the goal of stabilizing global warming to well below 2 °C[65]. We consider three decarbonization scenarios with different assumptions on technology availability. In the default FullTech scenario, no technology constraints are applied, such that the cost-minimizing mix of low-carbon supply options is chosen. To contrast the implications of mitigation strategies with opposing visions of the future role of variable renewable electricity generation, we considered two technology variants Conv and NewRE. In the Conv scenario, the share of variable renewable electricity supply is limited to 10%, resulting in an energy system largely based on conventional thermal power plants, with a strong emphasis on nuclear and CCS. In the NewRE scenario, by contrast, CCS is assumed to be unavailable and nuclear power is phased out, resulting in a scenario with large shares of electricity supply from new renewables, i.e., wind and solar technologies. The IAM scenarios are available as Supplementary Data 1. Similar sensitivity cases were considered in earlier IAM studies assessing energy system and cost implications of technology choice[5,6,36].

All IAMs used here represent the continued evolution of energy systems and technological progress in energy technologies over time, either by applying exogenous cost reductions based on bottom-up estimations (GCAM and MESSAGE models), or through endogenous modeling of learning-by-doing (REMIND, IMAGE, POLES models). Regional differences in renewable energy deployment are determined by regional resource potentials[66,67], cost and availability of competing technologies[36], and temporal matching between RE supply and demand[64,68]. The IAMs do not account for the impact of climate change on renewable energy resources. Our analysis also accounts explicitly for the deployment of storage and the expansion of long-distance transmission grids to sustain a stable power supply at high shares of wind and solar. We estimate short-term storage requirements, calculated based on the values in ref. [64]: For eight world regions, storage deployment is optimized by the hourly dispatch and investment model DIMES for a wide range of wind and solar shares. From these results, a polynomial fit is derived. In the current study, we apply this equation to the wind and solar energy deployed by the five IAMs. By default, storage is assumed to be deployed by default as lithium-ion batteries, while compressed-air and pumped-hydropower storage systems are considered as sensitivity cases. Long-distance transmission grid expansion is estimated based on a generalized equation derived from scenarios produced with the hourly dispatch and investment model REMIX with endogenous transmission grid expansion[48], from which the additional grid investment per unit of wind and solar energy is calculated as a function of the share of wind and solar in total power supply.

**Life-cycle assessment modeling.** Life-cycle impact assessment (LCIA) methods encompass models and characterization factors that aim at converting a list of environmental interventions into indicators representing environmental impact categories. The variety of impact assessment models available for LCA is wide, and the uncertainty attached with each set of characterization factors may also range widely. Additionally, most impact categories can be defined at two levels: midpoint, representative of the actual environmental phenomena caused by the life-cycle system, and endpoint, reflecting actual damage on areas of protection (human health, ecosystems, resources). The latter level brings more information relevant to a tradeoff to bear, but also introduces modeling uncertainty[69]. Due to these various layers of calculation, the final results of an LCIA also combine layers of uncertainties, usually grouped under three types[70]: parameter uncertainty (arising from the imprecision or incompleteness of input data, e.g. some substances are absent from the inventory or not characterized), model uncertainty (arising from inaccurate model representations of actual environmental phenomena, e.g. assuming the linearity of dose−response relationships), or value choices (e.g., setting a shorter or longer time horizon, depending on the perspective the practitioner wants to adopt). Parameter uncertainty is ontological and cannot be robustly estimated (one does not know what is not known). Model uncertainty can be estimated to the extent that natural phenomena can actually be modeled (which is not the case for all indicators). Uncertainty factors are not available in the ReCiPe impact assessment method used here[23]. Among three sets of characterization factors reflecting different value choices available from ReCiPe, we chose the hierarchical perspective, a middle-of-the-road approach whereby impacts are calculated for a 100-year time horizon.

For these reasons, impact uncertainty has not been assessed nor represented in the current results. The main limitations regarding impact assessment in this work are: the set of characterization factors used in this work reflect European conditions[23] (i.e., they are not spatially explicit, and not necessarily representative for the global scale); and the coverage of impact pathways and impact categories is incomplete due to gaps in the availability of characterization factors during data analysis for this work (e.g., our analysis does not consider ocean acidification impact[71,72], contributions of rare earth elements[57] or phosphorus to mineral resource depletion, or contributions from marine eutrophication to ecosystem

damage). Continuous improvements are being brought to identify, quantify and reduce LCIA uncertainty. Recent efforts include e.g. spatializing characterization factors with consideration of both species-richness and species vulnerability information[73–75], separating uncertainty from variability[76], or developing more significant indicators[77]. The UNEP-SETAC consensus initiatives aim at streamlining use of LCIA methodologies, as well as addressing known issues with the limitations that have been listed above[78–82]. The set of indicators and their respective characterization factors, as recommended by the UNEP-SETAC consortium was not published as a unified set at the time of modeling in this present work.

The life-cycle impact coefficients used for this method were derived from the integrated LCA model THEMIS (Technology Hybridized Environmental-economic Model with Integrated Scenarios)[9,24]. THEMIS is multiregional, representing nine world regions, and prospective, i.e., it accounts for future changes in technology performances along two future economic and technological storylines, one of which represents baseline developments, while the other follows stringent climate change mitigation strategies based on the IEA BLUE Map scenario[83]. THEMIS integrates the life-cycle inventory database ecoinvent 2.2 [22,84], which has been adapted to represent selected regional and future technology characteristics, and extended with extra electricity technology inventories unavailable in the original database. The version used in this analysis was augmented compared to the original version of THEMIS to also include nuclear power and biomass with and without CCS. It now includes 11 technology groups (photovoltaics, concentrated solar power, coal without/with CCS, natural gas without/with CCS, hydropower, wind power, nuclear power, biopower without/with CCS) represented by 43 different systems with their own life-cycle inventories[26,31,85].

The adaptations to ecoinvent consist of two main elements. First, the electricity mixes were changed according to the generic IEA Baseline and BLUE Map climate policy scenarios[83] for the years 2010, 2030 and 2050. Second, industrial energy efficiency and emission intensity improvements were modeled for nine major material production industries: aluminum, copper, ferronickel, nickel, iron, silicon, zinc, clinker, and flat glass—following the results of the New Energy Externalities Developments for Sustainability (NEEDS) project[86]. As a result, material production in 2050 under the climate change mitigation storyline benefits from lower energy fuel requirements, cleaner electricity and reduced emission intensities compared with material production in 2010.

For the characterization of biopower, THEMIS was augmented with bioenergy-induced land-use changes, $CO_2$, $NO_x$ and $CH_4$ emissions, as well as fertilizer requirements derived from a set of scenarios from the MAgPIE land-use model[63,87]. This approach allows to fully account for indirect land-use changes and emissions, e.g. from the relocation of nonenergy crops in response to increasing bioenergy demands. Land-use impacts from bioenergy depend critically on the assumptions regarding land management practices and policies. We therefore incorporate in THEMIS a variety of nine different scenarios from the MAgPIE model, as described in the following section. In MAgPIE, residues available for bioenergy use are exogenously assumed to amount to 47 EJ in 2050, corresponding to around one third of total global bioenergy demand projected in the IAM climate change mitigation scenarios for 2050. The MAgPIE scenarios assumed that additional 100 EJ are obtained from purpose-grown bioenergy production. For a further description of the implementation of MAgPIE scenarios into THEMIS, see the supplementary material of ref. [26].

The LCA coefficients derived from THEMIS are available as Supplementary Data 2.

**Integration of IAM scenarios and LCA.** Table 2 provides an overview of the environmental impacts considered at the midpoint of the cause-effect-chain and brief descriptions of the approaches used to estimate impact indicator results. A central strength of our study is the integration of energy flows and environmental impacts represented directly by IAMs and prospective LCA coefficients derived from the THEMIS model[26]. Following recommendations of ref. [26], we combine LCA energy coefficients broken down into life-cycle phases (i.e., construction, operation and end-of-life phases for each power-generation technology) and energy carriers (solid, liquid and gaseous fuels, and electricity) with IAM scenario-specific data (including emission intensities of electricity and other energy carriers, load factors of power plants, and power-generation capacity and generation), in order to estimate direct and indirect emissions associated with combustible energy carriers. ReCiPe impact assessment methods[23] are then used to characterize and aggregate the effects of these emissions into impact scores for particulate matter formation, photochemical oxidant formation, and terrestrial acidification. For toxicity impacts and several other impact types caused by environmental stressors or mechanisms not commonly associated with combustion and also not represented in IAMs (e.g., releases of radioactive or toxic substances affecting human health, or eutrophying substances harmful to ecosystems), we use the same approach except that we rely more on data (e.g., emission factors, mineral resource use) specified in the LCA and less on IAM model data. All coefficients obtained from THEMIS cover not only direct impacts, but also indirect impacts associated with mining, manufacturing and other supply chain activities. Geological $CO_2$ storage requirements are derived directly from the IAMs, as are water withdrawals connected to thermal power generation. Water withdrawal for irrigating bioenergy crops are obtained from the MAgPIE model.

**Table 2 Overview of methodologies for specific environmental impacts**

| Impact | Methodology |
|---|---|
| Human toxicity, freshwater ecotoxicity, marine ecotoxicity, terrestrial ecotoxicity, freshwater eutrophication, marine eutrophication, ionizing radiation, mineral resource depletion | Life-cycle impact coefficients derived from THEMIS for individual life-cycle stages of each power-generation option[26] are combined with activity data (i.e., new installed capacities, power generation) from the IAM scenarios. For the assessment of bioenergy, region- and scenario-specific yield ratios, nitrogen and phosphorus fertilizer requirements, and irrigation requirements dynamically derived from the MAgPIE model are incorporated into THEMIS. Effects of individual pollution and natural resource types are aggregated using ReCiPe characterization factors. |
| Land occupation and natural land transformation | Land occupation and transformation related to bioenergy crops are determined by MAgPIE. Bioenergy is assumed to be tradable. To account for indirect inter-regional (relocation of nonenergy crops and deforestation induced by bioenergy production) and intertemporal effects (conversion from forest to cropland facilitating bioenergy production at a later point in time), land transformation impacts from bioenergy were averaged across world regions and over the 2010−2050 period. Other direct and indirect land occupation and transformation associated with power-generation options are derived from THEMIS. |
| Particulate matter formation, photochemical oxidant formation, and terrestrial acidification | Air pollution emissions in IAM scenarios are based on technology-specific emission factors from the GAINS model (refs. [40,91]) combined with life-cycle coefficients of indirect energy requirements derived from THEMIS for individual life-cycle stages of each power-generation option[26] Air pollution emissions of $SO_2$, $NO_x$, $CH_4$, BC, OC and NMVOC are represented by source and energy technology in all IAMs. Effects of direct and indirect emissions of all pollution types are aggregated using characterization factors from ReCiPe[23]. |
| Water withdrawal | Water withdrawals for power plants (mostly cooling) are represented by source in all participating IAMs[11-13,51,92]. In addition, we account for irrigation of bioenergy as derived from the MAgPIE model. |
| Fossil resource depletion and geological $CO_2$ storage requirements | Fossil resource use and $CO_2$ storage requirements are represented explicitly in all IAMs. Also upstream fossil resource and $CO_2$ storage requirements are derived via combination with life-cycle coefficients of indirect energy requirements derived from THEMIS. |

Our analysis focuses on nonclimate environmental impacts of alternative climate change mitigation strategies, and therefore deliberately left out the climate change midpoint indicator.

The LCA coefficients are differentiated by two generic scenarios, indicating either a continuation of current trends (Baseline), or strong improvements in material and energy intensity of industrial processes (BLUE Map). These are matched to the IAM scenarios and regions as follows. The BLUE Map LCA coefficients are used for all IAM scenarios with stringent climate protection policies, whereas the Baseline LCA coefficients are used for IAM scenarios with no or insufficient mitigation efforts. IAM regions are matched to the THEMIS region with the best regional fit. The power systems' environmental impacts were calculated for each IAM model region, scenario and technology by multiplying the capacity additions and operation as derived from the IAMs with LCA impact coefficients derived with THEMIS and then aggregated to the global totals shown in the analysis of the paper.

Our combined IAM and LCA analysis captures the different timing of infrastructure and operational effects, allowing us to apply LCA impact data (e.g., emission factors) to the appropriate years when activities occur. It also allows us to capture the need to make upfront infrastructure investments for electricity generation later, the relevance of which has been noted by others[88,89]. Further, by decomposing LCA coefficients into four energy carriers (solid, liquid and gaseous fuels, and electricity) and assigning them to IAM scenario-specific emission intensities (for fuels and electricity), we are able to consistently account impacts related to indirect energy requirements[26].

Special attention is needed for environmental impacts related to land-use, since for many technologies, the definition of land occupation is highly uncertain or difficult to define, indirect effects (e.g., relocation of nonenergy croplands in response to bioenergy cultivation) play an important role, and while they are a dominant contributor to ecosystem damage, these damages are highly location-specific and therefore very uncertain in the global aggregate.

For biomass, land-use impacts depend critically on the assumptions regarding land management practices and policies. For purpose grown bioenergy, we therefore analyze nine cases with different assumptions on land management and policies (types of biomass feedstocks, biomass irrigation, regulation of land-use $CO_2$ emissions). Most importantly, we assume different stringency levels for the regulation of GHG emissions from the land-use sector, distinguishing three cases: No emissions regulation, weak regulation of agricultural and land-use change emissions, emulated using a carbon price of 5 $/tCO_2$ in 2020 increasing at 5%p.a.,

and strong regulation of agricultural and land-use change emissions, emulated using a carbon price of 30 $/tCO_2$ in 2020 increasing at 5% p.a., a level comparable to the $CO_2$ price in the energy sector required for the 2 °C limit. These sensitivity cases give rise to a substantial range in environmental impacts of bioenergy, such as land occupation and land transformation (see Supplementary Fig. 3). In particular, strong carbon regulation dis-incentivizes natural land transformation, and instead results in greater agricultural intensification, while weak carbon regulation exacerbates land-use impacts. The resulting per-unit bioenergy coefficients are documented in Supplementary Data 3.

The results on land occupation for nonbioenergy technologies are also highly uncertain. For instance, the land occupation of PV depends crucially on the share of ground-mounted vs. buildings-integrated solar, which we here assume to reach 75% by 2050. With exceptions[90], onshore wind power LCA studies typically account for the direct footprint of the wind turbine, machine houses and access roads, but not the space between wind turbines in a wind park, which can remain natural habitat or be exploitable for other uses, e.g. agriculture or forestry[9,29]. The total land occupied by an onshore wind farm is around 50−200 m²a/MWh, much higher than the direct wind infrastructure space requirement[26].

The aggregation of midpoint environmental impacts to the three endpoint impacts human health, ecosystem damage and resource depletion presented in Fig. 6 is also based on the ReCiPe methodology[23]. No midpoint-to-endpoint characterization is available in ReCiPe 2008 for marine eutrophication and water withdrawals (owing to unavailability of adequate assessment methods) and geological $CO_2$ storage (not considered in ReCiPe).

Resulting technology, scenario and IAM-specific environmental impacts are documented in Supplementary Data 4.

## Data availability

The datasets generated and analyzed during this study are available as supplementary data.

## Code availability

The code for integrating IAM scenario results with LCA environmental impact coefficients is available at Zenodo repository https://doi.org/10.5281/zenodo.3529760. The code also includes the analysis routines for creating the figures.

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

## Acknowledgements

The research leading to these results has received funding from the European Union's Seventh Programme FP7/2007-2013 under grant agreement no. 308329 (ADVANCE). Additional financial support came from the Research Council of Norway (grant 209697 for A.A. and E.G.H., and grant 288047 for A.A., G.L., F.H. and A.P.), and the Luxembourg National Research Fund for T.G. (grant INTER/RCN/18/12772313), the European Union's Horizon 2020 research and innovation program under grant agreement no. 730403 (INNOPATHS) for R.C.P., G.L. and M.P., as well as grant agreement no. 730035 (REINVENT) for H.S.d.B. and D.v.V. We would like to thank Sebastian Rauner for valuable discussions and helpful feedback.

## Author contributions

G.L., E.G.H., A.A. and M.P. designed the research and scenarios; A.A., E.G.H. and T.G. performed LCA modeling; M.P., H.S.d.B., O.F., M.H., G.I., I.M., R.C.P., S.M., M.v.d.B., D.v.V. performed IAM scenario modeling work; F.H., B.L.B. and A.P. performed land-use modeling of bioenergy impacts; R.C.P. contributed grid and storage modeling; M.P., G.L. and A.A. performed IAM/LCA integration; G.L. and M.P. performed data analysis and prepared all figures; G.L. wrote the paper with input from all authors.

## Competing interests

The authors declare no competing interests.
