## [Peer Review File · Nature Communications]

Reviewers' comments:

Reviewer #1 (Remarks to the Author):

The authors have proposed to evaluate the non-climate environmental impacts of deep decarbonization pathways using an integrated IAM and LCA approach. The modeling complexity to achieve this goal is grand and the study scope is immense. Regarding the novelty the authors indicate "there is only very limited system-level research on the environmental benefits and adverse side-effects of future decarbonized power supply." Much research, however, has already been conducted on this topic some of which has been acknowledged by the authors. The scale of this analysis is, however, novel.

Many of the findings from this paper are not unexpected given existing knowledge about high renewable penetrations. The authors confirm that renewables including wind and solar will reduce non-GHG pollutant emissions and related health effects but that biomass energy is more land intensive.

The scope of this modeling effort is definitely novel but there are many concerns about the value it offers given the scenarios used and uncertainty in modeling. The paper needs to provide a better overview of the scenarios and modeling approach and related limitations. In addition, there may be significant fundamental problems with the modeling.

Several major concerns include:

It is not clear if changes in technology (renewables, energy storage, infrastructure) over a 40 year period are modeled.

How are regional differences in deployment of the renewable technologies modeled.

How are changes in material requirements for these technologies over the next 30 years?

The limitations of the LCA impact assessment methods are not discussed. There are large uncertainties from modeling resource depletion and human and exotoxicity impacts. How are regional differences in impact assessment parameters treated? How are data gaps in addressed? The ReCiPe endpoint indicator used to compare fossil fuel and mineral resource depletion impacts may not represent future depletion surplus energy well.

How are climate change impacts in a base case heavy fossil with low renewable deployment modeled. I would expect that the ecosystem impacts from sea level raise and coastal flooding accelerated by more fossil fuel combustion to be significant. Biomass systems are land intensive but projections of coastal flooding are significant. It is not clear how this was modeled.

Climate change and technology feedbacks are not well discussed.

A scenarios data file is provided but it is not clear how the global data were constructed and synthesized from regional data.

Reviewer #2 (Remarks to the Author):

I have read the article "Environmental co-benefits and adverse side-effects of alternative power sector decarbonization strategies", which analysis the potential impacts of different future power generation technology mixes on a set of life-cycle indicators. By combining scenarios from integrated assessment models with methods from life-cycle assessment, the authors show that an

ambitious decarbonization of the power sector does not only reduce GHG emissions, but is also expected to reduce the sector's negative side-effects, while potentially increasing pressures on other indicators. Different technology portfolios are associated with different trade-offs in terms of positive and negative externality changes.

The combination of life-cycle analysis with integrated assessment modelling is a relatively recent and innovative research approach, the relevance of which is increasingly recognized, and has led to some very interesting results over the last few years - partly by the authors of this article (e.g., Hertwich et al. 2015, Gibon et al. 2017, Pehl et al. 2017). The analysis and the claims made in the article are thus not fundamentally novel as such, as they largely built on previous similar work. However, in my opinion the article goes beyond these existing studies in important ways, which will likely make it interesting for a wide readership. The major novelty is the integration of life-cycle analysis with scenarios from five integrated assessment models, while previous studies rely on one model. Furthermore, the authors do indeed provide a very comprehensive overview of a wide range of impacts, which will be an important contribution to the field, and which is highly relevant for current policy-making.

Generally, I find that the methodology is sound and robust, and the manuscript is well written. Still, I think that the article could benefit from some improvements. My major comments are:

1. Integration of results from different IAMs: It did not become clear to me how the authors combined the scenario results from their different models, which is arguably the main selling point here. For example, in the figure captions, authors state that "boxplots indicate (...) ranges across regions, technology variants and participating integrated assessment models". What is this supposed to mean exactly? Given that five models should result in five observations per scenario, how are interquartile ranges derived for such a small sample? Given that the figures present global aggregates, how do ranges across regions matter?

2. Analysis of biomass as a power technology: The results invoke the impression that in case of decarbonization scenarios, biomass use will be of major relevance for future side-effects of the power sector, at least on several indicators. However, the presentation of main results in absolute terms (instead of relative impacts per MWh) do not make sufficiently transparent what the relative side-effects of biomass are, compared to other technologies. I suggest that relative impacts per MWh are discussed more explicitly, and not only presented in the SI. Considering the discussion about biomass and BECCS in the literature, it would also be interesting to know in how far larger shares of biomass use would generally lead to larger adverse side-effects?

3. Time scale considered: Why is the analysis restricted to the timeframe until 2050, although all of the included integrated assessment models are normally run until 2100? For example, this could be of large relevance for evaluating the long-term impacts of biomass as a power technology: IAM scenarios reviewed in the IPCC reports (AR5 and SR15) generally show a pattern in which the use of biomass becomes much larger towards end-century, compared to still relatively low levels in 2050. Given the large side-effects from biomass, this should at least be discussed. Also, which timeframe is considered for side-effects in general, given that they could also occur after 2050?

In addition, I have some minor comments:

Throughout: Authors should proof-read the manuscript, as it still contains various typos and editing errors.

Line 45: Double-check if reaching net emission neutrality by mid-century is indeed an explicitly stated aim of the Paris Agreement, or if it is merely an implication of the well-below 2C target.

Line 75: What about prospective LCA studies on energy end-use technologies, such as the studies by Cox et al. (2018) and Mendoza Bertran et al. (2018)?

Line 164: In case of nuclear, authors should consider the possibility of nuclear accidents much earlier, and not just in the discussion. Clearly, simply looking at business-as-usual operation is a bit naïve for this technology. What are the impacts from incidents such as Fukushima, and could they be factored in based on their probabilities of occurring, maybe as a sensitivity analysis?

Line 206: How does the pressure on land from biomass use depend on the assumptions regarding the sources of biomass? For example, to which extent is land conversion prevented by model assumptions on the availability of residues?

Line 213: What are the relative variabilities which stem from models, scenarios and assumed management practices?

Figure 3: How is the technology mix in the outlier decarbonization scenarios different from the mean scenario? Do they have significantly more biomass (or other technology) in their mix, for example? Do outliers result from the same models in all cases?

Lines 280-289: Storage capacities for CCS are another point for which the analysis until 2100 would be interesting, as storage requirements tend to further grow towards end-century.

Lines 290-305: How are resource demands from the construction phase accounted for? Completely in the year of construction, or distributed over the entire use-phase?

Line 347: What about a biomass and land-use strategy?

Line 360: Tradeoff between what?

Line 362: Reference needed. Do you mean CO₂ reductions in the power sector only?

Figure 5: Clarify units. I assume all values should be per year?

Line 392-402: What is the reasoning behind the specification of scenario assumptions, such as the limitation of renewables to 10%?

Line 429: Are the electricity mixes for the ecoinvent adaptation taken from the analyzed IAM scenarios, or were different scenarios used?

Table 2: What are the assumptions regarding the land allocation for bioenergy crops? Are crops grown in the same region where they are used, or globally allocated based on some optimality consideration?

Line 498: In how far are the assumed carbon prices for land-use consistent with the carbon prices for the energy sector in the decarbonization scenarios? Do authors imply that it is realistic that a carbon price would indeed be introduced on land-use, or do authors only use the price as a proxy for assumed regulations on land-use?

We would like to thank the Editor and the three Referees for their constructive comments. We address these point by point below, with our responses in **purple font**, and also marked the major changes to the manuscript in **purple font**.

Reviewer #1 (Remarks to the Author):

The authors have proposed to evaluate the non-climate environmental impacts of deep decarbonization pathways using an integrated IAM and LCA approach. The modeling complexity to achieve this goal is grand and the study scope is immense. Regarding the novelty the authors indicate "there is only very limited system-level research on the environmental benefits and adverse side-effects of future decarbonized power supply." Much research, however, has already been conducted on this topic some of which has been acknowledged by the authors. The scale of this analysis is, however, novel.

Many of the findings from this paper are not unexpected given existing knowledge about high renewable penetrations. The authors confirm that renewables including wind and solar will reduce non-GHG pollutant emissions and related health effects but that biomass energy is more land intensive.

The scope of this modeling effort is definitely novel but there are many concerns about the value it offers given the scenarios used and uncertainty in modeling. The paper needs to provide a better overview of the scenarios and modeling approach and related limitations. In addition, there may be significant fundamental problems with the modeling.

Thank you for this overall assessment! We appreciate the reviewer's appraisal of novelty. We reworked the presentation to improve the overview of the scenarios and methodology, as described in the point-by-point responses below.

1. *It is not clear if changes in technology (renewables, energy storage, infrastructure) over a 40 year period are modeled.*

All IAMs used here represent the continued evolution of energy systems and technological progress in energy technologies over time, either by applying exogenous cost reductions based on bottom-up estimations (GCAM and MESSAGE models), or through endogenous modeling of learning-by-doing (REMIND, IMAGE, POLES models). Increased deployment of variable renewable energy (VRE) renewable leads to new demands for energy storage and infrastructure, which is represented in the models, albeit at an aggregate scale. IMAGE, MESSAGE, POLES, and REMIND participated in the detailed analysis and improvement of their renewable energy modeling undertaken in a companion activity of the ADVANCE research project, which led to two in-depth comparison papers (Luderer et al. 2017; Pietzcker et al. 2017) as well as papers for each model, documenting the specific approach chosen in each model (linked in the Pietzcker et al. Paper). Question 3 below addresses how the effect of technology development on environmental impacts is represented. We added clarifying text to the methods section.

2. *How are regional differences in deployment of the renewable technologies modeled?*

Regional differences in renewable energy deployment are mostly a function of (1) regional resource potentials, (2) temporal matching between RE supply and demand, and (3) cost and availability of

competing technologies (e.g., nuclear, CCS). In the above-mentioned companion activity of the ADVANCE research project on variable renewable energy integration, new and refined regional resource potentials were developed (Pietzcker et al. 2014; Eurek et al. 2017), as well as new data on regional temporal patterns, specifically the diurnal and seasonal temporal matching between RE supply and demand (Pietzcker et al. 2017; Ueckerdt et al. 2017). Based on these regional differences of suitability of renewable resources and the differences in existing energy system structures and competing resource endowments, regional differences in renewable deployment emerge. We added clarifications to the methods section regarding regional deployment differences.

3. *How are changes in material requirements for these technologies over the next 30 years?*

Changes in material requirements are considered to some degree, depending on the technology. For all technologies, changes in power plant energy efficiencies, load factors and lifetimes in the scenarios, lead to changes in the material requirements per unit of electricity generated over the lifetime. For solar PV, we additionally account for significant increases in material efficiencies over time, due to a combination of lighter bill of materials and a gradual shift towards thin-film PV technologies (Bergesen et al. 2014) and the related discussion on PV in Arvesen et al. (2018). The special consideration of PV material efficiency is motivated by the particularly fast technological progress for PV technologies. For wind power, we recognize that material compositions and material efficiencies may change in various ways (Caduff et al. 2012; Hertwich et al. 2016; Mishnaevsky et al. 2017), but they are expected to be much smaller than for solar PV. The dominant trends, especially when moving into very large wind turbine sizes, are rather unclear and we wish to avoid making arbitrary assumptions. We have not been able to include material composition/efficiency changes for wind power. The same applies for solar CSP. We added clarifications in this respect to the methods section.

4. *The limitations of the LCA impact assessment methods are not discussed. There are large uncertainties from modeling resource depletion and human and ecotoxicity impacts. How are regional differences in impact assessment parameters treated? How are data gaps in addressed? The ReCiPe endpoint indicator used to compare fossil fuel and mineral resource depletion impacts may not represent future depletion surplus energy well.*

These are legitimate and relevant points/questions. We discussed imitations of LCA impact assessment methods in the last two paragraphs of Methods, LCA modeling. In order to add more emphasis on impact assessment limitations in our manuscript, we have moved the discussion from the Methods, LCA modeling section, to the *Combined assessment and policy conclusions* section.

I am afraid we do not understand specifically what you mean by “the ReCiPe endpoint indicator ... may not represent future depletion surplus energy well”.]

5. *How are climate change impacts in a base case heavy fossil with low renewable deployment modeled. I would expect that the ecosystem impacts from sea level raise and coastal flooding accelerated by more fossil fuel combustion to be significant. Biomass systems are land intensive but projections of coastal flooding are significant. It is not clear how this was modeled. Climate change and technology feedbacks are not well discussed.*

The main goals of the study was to quantify co-benefits and adverse side-effects of climate policies, and to compare non-climate environmental impacts of alternative energy transformation strategies

achieving the same climate target. The whole analysis focuses on these non-climate environmental impacts of alternative energy system configurations leading to 2°C-consistent greenhouse gas emissions reductions, while the impacts of climate change were deliberately left out of the analysis. The main comparison is between the three climate mitigation scenarios FullTech, Conv and NewRE, which all achieve the same climate target and will thus have comparative additional impacts from climate change. The comparison to the Base scenario without climate mitigation is intended to assess environmental and adverse side-effects of the energy system transformation required for climate change mitigation. Coastal flooding is not yet included in life-cycle impact assessment methodologies, and thus considering it in the analysis of the energy systems land footprint will have to be left to future research.

We added the statement *“The IAMs do not account for the impact of climate change on renewable energy resources”* to the methods section.

6. A scenarios data file is provided but it is not clear how the global data were constructed and synthesized from regional data.

The integrated assessment models participating in this study represent 10-12 world regions, as documented in the respective model documentations available at https://www.iamcdocumentation.eu/index.php/IAMC_wiki.

The regions of the IAM models were mapped to the nine world regions represented in the THEMIS LCA model. The power systems' environmental impacts were calculated for each IAM model region, scenario and technology by multiplying the capacity additions and operation as derived from the IAMs with LCA impact coefficients derived with THEMIS and then aggregated to the global totals shown in the analysis of the paper. We added further clarification to the methods section of the paper.

Reviewer #2 (Remarks to the Author):

I have read the article “Environmental co-benefits and adverse side-effects of alternative power sector decarbonization strategies”, which analysis the potential impacts of different future power generation technology mixes on a set of life-cycle indicators. By combining scenarios from integrated assessment models with methods from life-cycle assessment, the authors show that an ambitious decarbonization of the power sector does not only reduce GHG emissions, but is also expected to reduce the sector’s negative side-effects, while potentially increasing pressures on other indicators. Different technology portfolios are associated with different trade-offs in terms of positive and negative externality changes. The combination of life-cycle analysis with integrated assessment modelling is a relatively recent and innovative research approach, the relevance of which is increasingly recognized, and has led to some very interesting results over the last few years - partly by the authors of this article (e.g., Hertwich et al. 2015, Gibon et al. 2017, Pehl et al. 2017). The analysis and the claims made in the article are thus not fundamentally novel as such, as they largely built on previous similar work. However, in my opinion the article goes beyond these existing studies in important ways, which will likely make it interesting for a wide readership. The major novelty is the integration of life-cycle analysis with scenarios from five integrated assessment models, while previous studies rely on one model. Furthermore, the authors do indeed provide a very comprehensive overview of a wide range of impacts, which will be an important contribution to the field, and which is highly relevant for current policy-making.

Generally, I find that the methodology is sound and robust, and the manuscript is well written. Still, I think that the article could benefit from some improvements. My major comments are:

We appreciate the reviewer’s generally positive assessment.

- 1. Integration of results from different IAMs: It did not become clear to me how the authors combined the scenario results from their different models, which is arguably the main selling point here. For example, in the figure captions, authors state that “boxplots indicate (...) ranges across regions, technology variants and participating integrated assessment models”. What is this supposed to mean exactly? Given that five models should result in five observations per scenario, how are interquartile ranges derived for such a small sample? Given that the figures present global aggregates, how do ranges across regions matter?*

Thanks for pointing out the need for a clearer description of how scenarios were combined and uncertainties derived. Our IAM-LCA analysis system derives impacts for each model region, scenario and technology. THEMIS represents various variants for each technology class, e.g., various types of solar photovoltaic technologies, with default assumptions regarding relative market shares of these technologies in the future. The sums of regional results yield global aggregate results. Central estimates shown in the figures were derived by averaging across the participating models. To derive uncertainty estimates, we combined differences across technology variants with differences across IAM scenarios, typically resulting in 15-50 individual point estimates per technology. We added further explanation to the methods section, and clarified the meaning of uncertainties ranges in the caption.

The reviewer is right to point out that regional ranges are not meaningful for the global aggregates. In fact, this statement was inaccurate and is now removed.

- 2. Analysis of biomass as a power technology: The results invoke the impression that in case of decarbonization scenarios, biomass use will be of major relevance for future side-effects of the power sector, at least on several indicators. However, the presentation of main results in absolute terms (instead of relative impacts per MWh) do not make sufficiently transparent what the relative side-effects of biomass are, compared to other technologies. I suggest that relative impacts per MWh are discussed more explicitly, and not only presented in the SI. Considering the discussion about biomass and BECCS in the literature, it would also be interesting to know in how far larger shares of biomass use would generally lead to larger adverse side-effects?*

We agree that the per-MWh impacts are highly relevant. As suggested by the reviewer, we moved a condensed version of Supplementary Figure S2 to the main text.

Our estimates rely on marginal impacts at a 100 EJ bioenergy production level in 2050. Since the life-cycle impact assessment methodology used here relies on a linearization of the relation between stressors and impacts, we cannot make statements to what extent marginal impacts increase with increasing bioenergy demands. This needs to be left to further research.

- 3. Time scale considered: Why is the analysis restricted to the timeframe until 2050, although all of the included integrated assessment models are normally run until 2100? For example, this could be of large relevance for evaluating the long-term impacts of biomass as a power technology: IAM scenarios reviewed in the IPCC reports (AR5 and SR15) generally show a pattern in which the use of biomass becomes much larger towards end-century, compared to still relatively low levels in 2050. Given the large side-effects from biomass, this should at least be discussed. Also, which timeframe is considered for side-effects in general, given that they could also occur after 2050?*

Indeed, the IAM scenarios underlying this study (as well as most others documented in the literature) run out to 2100. Three of the models (IMAGE, MESSGE, REMIND) also reported results for the entire century, while the other two (GCAM, POLES), in view of the predefined temporal scope of the study, only reported results until 2050 to the database.

We focused our analysis on the time frame until 2050 for two reasons: Firstly, the transformation of the power system is almost completely accomplished already by mid-century, with 93-98% of electricity generation originating from non-fossil or CCS capacities. Secondly, prospective life-cycle assessment with THEMIS is only available until 2050. The methodology requires detailed technology assumptions at a level of granularity that goes beyond the detail provided by IAMs (e.g., regarding the relative shares of different PV technologies), which are difficult to project beyond 2050.

Importantly, in line with standard practice in LCIA, impacts reported encompass all effects induced by activities in the year under consideration, including those occurring at a later point in time. This is particularly relevant for toxicity impacts from resource extraction: Leeching of toxic substances from mine dumps is a slow process that occurs over decades or even centuries. Our analysis takes these long-term impacts into account.

In addition, I have some minor comments:

4. *Throughout: Authors should proof-read the manuscript, as it still contains various typos and editing errors.*

Thank you for pointing this out. We proof-read and edited the entire manuscript.

5. *Line 45: Double-check if reaching net emission neutrality by mid-century is indeed an explicitly stated aim of the Paris Agreement, or if it is merely an implication of the well-below 2C target.*

In its Article 4, the Paris Agreement explicitly states “In order to achieve the long-term temperature goal set out in Article 2, Parties aim ... to undertake rapid reductions thereafter in accordance with best available science, so as to achieve a balance between anthropogenic emissions by sources and removals by sinks of greenhouse gases in the second half of this century...”. We therefore kept the statement about GHG neutrality in the introduction.

6. *Line 75: What about prospective LCA studies on energy end-use technologies, such as the studies by Cox et al. (2018) and Mendoza Bertran et al. (2018)?*

We pointed out literature on end-use technologies, and added these two references

7. *Line 164: In case of nuclear, authors should consider the possibility of nuclear accidents much earlier, and not just in the discussion. Clearly, simply looking at business-as-usual operation is a bit naïve for this technology. What are the impacts from incidents such as Fukushima, and could they be factored in based on their probabilities of occurring, maybe as a sensitivity analysis?*

LCA, by tradition and design, is separate from risk assessment. Hofstetter et al. (2002) discussed the purposes of various comparative assessment methods, including LCA and various types of risk analysis. They stated that LCA “includes only accidents that occur frequently enough that their emissions are included in yearly statistical compilations”. In recent years, there have been attempts to integrate the consequences of accidents into LCA (Burgherr et al. 2012; Hirschberg et al. 2016; Volkart et al. 2016, 2017). We welcome these attempts, but, at the same time, we have not been able to consider accidents in the current study. Additionally, providing accident information would need to be done for all power generation alternatives, not only nuclear. This could be done, as data is available for most electricity production chains; recent literature (Hirschberg et al. 2016) suggests that severe accidents of nuclear power plant operations cause fewer deaths (years of life lost, YOLLs) per GWh than fossil fuels, and as much as wind offshore or combined heat and power from biomass. Based on current literature, it is therefore not clear that consistently including a risk assessment approach in LCA would result in higher impacts for nuclear power relative to other fossil fuel power generation options.

8. *Line 206: How does the pressure on land from biomass use depend on the assumptions regarding the sources of biomass? For example, to which extent is land conversion prevented by model assumptions on the availability of residues?*

The land-related environmental impacts depend significantly on feedstocks. We used sensitivity cases on the bioenergy sources regarding irrigation of energy crops, use of woody or woody and grassy bioenergy feedstocks. These add to the uncertainty ranges presented in the bioenergy results.

Regarding residues, MAGPIE assumed to these to amount to 50 EJ in 2050 and not to compete with crop production and forestry. This amount compares to a total primary bioenergy demand in that year of 150 EJ averaged across mitigation scenarios. We added a clarification on residues assumptions to the methods section.

9. *Line 213: What are the relative variabilities which stem from models, scenarios and assumed management practices?*

We added a new Suppl. Figure S4 on land occupation and natural land transformation impacts broken down by land use assumptions. Management practices and land use policy frameworks as varied in the different land-use modeling assumptions are the major contributor to overall uncertainty, resulting in around a factor of 10 difference between the cases with lowest and highest impacts. We also find the differences in land impacts across models to be of similar magnitude. We also rephrased the last sentence of the abstract to emphasize the uncertainty in ecosystem damages.

10. *Figure 3: How is the technology mix in the outlier decarbonization scenarios different from the mean scenario? Do they have significantly more biomass (or other technology) in their mix, for example? Do outliers result from the same models in all cases?*

Related to point 6. above. The very high outliers in the case of land occupation and natural land transformation stem from the combination of scenario realizations with high biomass input (70-105 EJ in the POLES model) in combination with pessimistic assumptions about land use policy and land – intensive management (no carbon pricing in the land sector, no irrigation, woody bioenergy feedstocks).

11. *Lines 280-289: Storage capacities for CCS are another point for which the analysis until 2100 would be interesting, as storage requirements tend to further grow towards end-century.*

We agree on the relevance of longer time horizons for CCS, and added a corresponding plot to the supplementary materials. We added an analysis of CCS sequestration rates out to 2100 for the IMAGE, MESSAGE and REMIND models (due to the predefined scope of the analysis, POLES and GCAM reported results only until 2050 to the common database, and therefore are not shown). CCS deployment continues to increase in the IMAGE and MESSAGE LimVRE scenarios during 2nd half of the century. In REMIND LimVRE and MESSAGE FullTech, power sector CCS decreases after 2050 due to competition for CCS storage capacities with non-electric CCS applications (industry CCS, biomass-to-H₂, biomass-to-liquids).

12. *Lines 290-305: How are resource demands from the construction phase accounted for? Completely in the year of construction, or distributed over the entire use-phase?*

We attributed the resource demands for construction entirely to the year of construction, not the use phase. Now clarified in the text.

13. *Line 347: What about a biomass and land-use strategy?*

We agree that an integrated and coordinated biomass and land-use strategy is crucial, and added that point to the discussion of biomass (2nd to last paragraph): *“This underlines the need for an*

integrated global land management to overcome the tradeoff between climate change mitigation and conservation.”

14. *Line 360: Tradeoff between what?*

Reformulated, see response to point 13 above.

15. *Line 362: Reference needed. Do you mean CO2 reductions in the power sector only?*

The power supply has the greatest potential for CO2 reductions until 2030, and wind and solar are greatest contributors to power sector decarbonization until 2030, as recent projections show.

Clarified, and added Vrontisi et al. (2018) as a recent highly relevant reference.

16. *Figure 5: Clarify units. I assume all values should be per year?*

Thanks for spotting this. Indeed units are per year, and were **corrected accordingly**.

17. *Line 392-402: What is the reasoning behind the specification of scenario assumptions, such as the limitation of renewables to 10%?*

The idea behind the two limitation scenarios NewRE and Conv is to contrast the implications of mitigation strategies with opposing visions of the future role of variable renewable electricity generation. The exact specification of these bounding cases is inevitably somewhat arbitrary, but follows prevalent perspectives in the scientific discourse.

There has been slow progress and cost overruns with new nuclear constructions, as well as a number of nuclear expansion moratoria and even nuclear phase-out policies in some countries. Similarly, CCS deployment faces major setbacks. Against this background, several scholars have argued for “all renewables” decarbonization strategies, which are emulated in the “NewRE” scenario. On the other end of the spectrum, some scholars have argued that the potential wind and solar energy is limited due to variability of their supply. This perspective is reflected in the Conv scenario, in which the share of wind and solar electricity supply was limited to 10% of total electricity – implying a 35% increase from current levels. We also could have chosen a somewhat lower or higher limit, but expect that a choice would not have changed the results and insights from the study in significantly.

Similarly, we developed a scenario that mostly focused on nuclear and CCS, which required setting bounds on the wind/solar share in the models. Fixing it to 0 felt like a too unrealistic scenario – integration challenges are so low at low wind/solar shares that it is difficult to imagine a country completely banning wind and solar power. On the other hand, at higher shares integration challenges start to become more relevant, and public opposition often increases with increased deployment, so we set a limit to 10%.

18. *Line 429: Are the electricity mixes for the ecoinvent adaptation taken from the analyzed IAM scenarios, or were different scenarios used?*

The ecoinvent adaptation was based on the IEA Reference / BLUE Map scenarios (addressing electricity mixes and energy/material efficiency improvements for key industries), as this is the only adaptation currently available in THEMIS. In order to make this clearer, we adjusted the text in the Methods, LCA modeling. In relation to this, note that we integrate life cycle energy flows (including electricity flows) derived from the LCA with IAM scenario-specific emission intensities, as explained in

Methods, Integration of IAM scenarios and LCA. This means that in practice, emissions associated with electricity are largely consistent with the IAM scenarios.

19. Table 2: What are the assumptions regarding the land allocation for bioenergy crops? Are crops grown in the same region where they are used, or globally allocated based on some optimality consideration?

Models generally assume bioenergy to be tradable, so they are not necessarily grown in the same region where they are used. By using the landuse management model MAgPIE, which also account for international trade in crops, we can account for not only for domestic but also foreign impacts of bioenergy cultivation on cropland demand (indirect landuse change. Clarified in the table.

20. Line 498: In how far are the assumed carbon prices for land-use consistent with the carbon prices for the energy sector in the decarbonization scenarios? Do authors imply that it is realistic that a carbon price would indeed be introduced on land-use, or do authors only use the price as a proxy for assumed regulations on land-use?

Since in the real world emissions regulations of energy and land use systems are very different, we deliberately deviate from the assumption that carbon prices in the energy and land sectors are harmonized. Instead, we take climate policy stringency in the land use sector as an uncertainty dimension and use carbon prices as a proxy to emulate different levels of stringency of the regulation: (1) No emissions regulation in the land sector, (2) weak regulation of agricultural and landuse change emissions, emulated using a carbon price of 5\$/tCO₂ in 2020 increasing at 5%p.a., and (3) strong regulation of agricultural and landuse change emissions, emulated using a carbon price of 30 \$/tCO₂ in 2020 increasing at 5%p.a., a level comparable to the CO₂ price in the energy sector required for the 2°C limit. We added more clarification to the text.

Reviewers' comments:

Reviewer #1 (Remarks to the Author):

The paper has been improved but the authors need to address the following.

The authors addressed several of limitations of their study in the revision and also provided details regarding other questions raised in the review.

It still is necessary to provide more emphasis in the paper on the uncertainties and limitations associated with impact assessment methods used in this study. It would be useful to highlight in the methods section of the paper the types of uncertainties from data gaps, impact methods, and characterization factors and how these are represented in the uncertainty ranges given in the paper. The uncertainty description now provided in Combined assessment and policy conclusions section of the paper does not provide any references to uncertainty in LCIA; such references should be added. There are the large uncertainties in human toxicity, ecotoxicity and resource depletion LCIA results. This background on uncertainty needs to be added particularly given that many of the Nature Comm readers will not be familiar with the LCA field. Also the error bars given for results need to be described in this context.

Regarding comment 5 and the response: Coastal flooding is not yet included in life-cycle impact assessment methodologies, and thus considering it in the analysis of the energy systems land footprint will have to be left to future research. "The IAMs do not account for the impact of climate change on renewable energy resources" to the methods section. This does not address the issue that the current baseline of fossil energy use is contributing to coastal flooding and land use changes. The land intensity of bioenergy is highlighted in this study and compared against land requirements for the current system (heavily fossil) but the current system is responsible for land use changes not quantified in LCIA methods which you point out above. Even though the focus is on non-climate impacts and that coastal flooding is not modeled in LCIA you should state this in the paper. The reader will compare land use requirements for biomass and other renewable with the base case, which is responsible for land use changes (coastal flooding) not included in your study.

Reviewer #2 (Remarks to the Author):

The authors have thoroughly addressed my earlier comments. In general, the revised manuscript has significantly improved in its quality. However, some concerns remain. My major comments regarding the revised article are:

1. Marginal impacts of bioenergy: In their reply to my earlier comment 2, the authors state that their results depend on marginal impacts at 100 EJ bioenergy production in 2050. In their reply to comment 18, however, they refer to 150 EJ as the primary bioenergy demand in 2050, averaged across mitigation scenarios. This raises the question in how far the estimated marginal impact of bioenergy production is consistent with the demand as projected by the IAMs. For various reasons (such as limited availability of suitable land), the marginal impacts per EJ could potentially be much larger at 150 EJ per year, compared to the used estimates at 100 EJ per year. If there is indeed such a large inconsistency between LCA estimates and IAM projections, this potentially constitutes a major weakness of the presented results for bioenergy. Ideally, the authors should base their calculations on the estimated marginal impacts which correspond to the projected level of bioenergy use in any year. If such estimates are not available, at the very least, such a caveat should be explicitly discussed.

2. Time scale considered: While it is certainly true that the decarbonisation of the electricity system would be largely completed by mid-century, this does not automatically imply that the

projected composition of the electricity system will remain constant thereafter. Looking at the new figure S5, BECCS seems to grow significantly in the second half of the century. Given the acknowledged large uncertainties surrounding bioenergy production, this could have major implications, in particular when also considering the potentially increasing marginal impact of bioenergy use at larger scales (see my comment above). This also needs to be made very explicit within the manuscript.

Some minor comments:

3. New Supplementary Figure S2: The new figure is much appreciated. The authors should add to the caption the explanation regarding the uncertainty ranges (which was also added to the other figures).

4. Integration of accidents: The clarification in the manuscript text is noted. I recommend to reference the mentioned literature which attempts to estimate such impacts.

5. l. 369: "the need for integrated global land management" is a rather unfortunate formulation. What is this supposed to look like in political reality? Perhaps replace by something along the lines of 'the need for integrated land management across the globe', or 'international policy frameworks for sustainable land management'?

6. l. 489: Part of the sentence is missing. 'Corresponding to' what?

We would like to thank the two Referees for their additional constructive comments. We address these point by point below, with our responses in purple font, and also marked the major changes to the manuscript in purple font.

Reviewer #1 (Remarks to the Author):

The paper has been improved but the authors need to address the following.

The authors addressed several of limitations of their study in the revision and also provided details regarding other questions raised in the review.

- 1. It still is necessary to provide more emphasis in the paper on the uncertainties and limitations associated with impact assessment methods used in this study. It would be useful to highlight in the methods section of the paper the types of uncertainties from data gaps, impact methods, and characterization factors and how these are represented in the uncertainty ranges given in the paper. The uncertainty description now provided in Combined assessment and policy conclusions section of the paper does not provide any references to uncertainty in LCIA; such references should be added. There are the large uncertainites in human toxicity, ecotoxicity and resource depletion LCIA results. This background on uncertainty needs to be added particularly given that many of the Nature Comm readers will not be familiar with the LCA field. Also the error bars given for results need to described in this context.*

Thanks for pointing to the need for a better reflection of uncertainties in impact assessments. In response, we

- Explained in the main text (ll. 114-166) and figure captions that ranges do not account for uncertainties in impact characterization;
 - Added an extended paragraph to the *Combined assessment* section discussing the uncertainties and limitations in impact assessment;
 - Added further discussion of and references on impact assessment uncertainty and state of research to the methods.
- 2. Regarding comment 5 and the response: Coastal flooding is not yet included in life-cycle impact assessment methodologies, and thus considering it in the analysis of the energy systems land footprint will have to be left to future research. "The IAMs do not account for the impact of climate change on renewable energy resources" to the methods section. This does not address the issue that the current baseline of fossil energy use is contributing to coastal flooding and land use changes. The land intensity of bioenergy is highlighted in this study and compared against land requirements for the current system (heavily fossil) but the current system is responsible for land use changes not quantified in LCIA methods which you point out above. Even though the focus is on non-climate impacts and that coastal flooding is not modeled in LCIA you should state this in the paper. The reader will compare land use requirements for biomass and other renewable with the base case, which is responsible for land use changes (coastal flooding) not included in your study.*

We included additional text to the discussion of land use impacts (around line 222 and following):

“In comparing fossil to non-fossil power generation, it is also important to emphasize that our analysis does not account for habitat losses caused by coastal flooding. Due to higher climate change induced sea level rise, coastal flooding will be more severe in the Base scenario than in the climate change mitigation scenarios.”

Reviewer #2 (Remarks to the Author):

The authors have thoroughly addressed my earlier comments. In general, the revised manuscript has significantly improved in its quality. However, some concerns remain. My major comments regarding the revised article are:

- 1. Marginal impacts of bioenergy: In their reply to my earlier comment 2, the authors state that their results depend on marginal impacts at 100 EJ bioenergy production in 2050. In their reply to comment 18, however, they refer to 150 EJ as the primary bioenergy demand in 2050, averaged across mitigation scenarios. This raises the question in how far the estimated marginal impact of bioenergy production is consistent with the demand as projected by the IAMs. For various reasons (such as limited availability of suitable land), the marginal impacts per EJ could potentially be much larger at 150 EJ per year, compared to the used estimates at 100 EJ per year. If there is indeed such a large inconsistency between LCA estimates and IAM projections, this potentially constitutes a major weakness of the presented results for bioenergy. Ideally, the authors should base their calculations on the estimated marginal impacts which correspond to the projected level of bioenergy use in any year.
If such estimates are not available, at the very least, such a caveat should be explicitly discussed.*

In fact, these two numbers are consistent, since the 150 EJ bioenergy demand refer to total bioenergy (purpose grown plus residues), while the MAgPIE results are based on 100 EJ purpose grown bioenergy production (excluding additional bioenergy from residues, which were assumed to account for another 47 EJ). Thanks for pointing out that this distinction was not clear from the previous version of the text. Reformulated in the Methods Section: “In MAgPIE, residues available for bioenergy use are exogenously assumed to amount to 47 EJ in 2050, corresponding to around one third of total global bioenergy demand projected in the IAM climate change mitigation scenarios for 2050. The MAgPIE scenarios assumed that additional 100 EJ are obtained from additional purpose-grown bioenergy production.”

- 2. Time scale considered: While it is certainly true that the decarbonisation of the electricity system would be largely completed by mid-century, this does not automatically imply that the projected composition of the electricity system will remain constant thereafter. Looking at the new figure S5, BECCS seems to grow significantly in the second half of the century. Given the acknowledged large uncertainties surrounding bioenergy production, this could have major implications, in particular when also considering the potentially increasing marginal impact of bioenergy use at larger scales (see my comment above). This also needs to be made very explicit within the manuscript.*

We added discussion of the choice of time scale and longer term impacts to the concluding section:

“We deliberately focused our analysis on the year 2050, since by mid-century the decarbonization of power system is largely completed and technology developments get increasingly uncertain with longer time horizons. Nonetheless, it is important to note that environmental impacts of a decarbonized might continue to evolve thereafter, depending on size and composition of power supply. For instance, increasing contributions of biomass to electricity generation, as projected in many IAM scenarios⁶³, will exacerbate ecosystem damages.”

Some minor comments:

3. *New Supplementary Figure S2: The new figure is much appreciated. The authors should add to the caption the explanation regarding the uncertainty ranges (which was also added to the other figures).*

Thanks. Added explanations on the uncertainty ranges to Figs. S2, S3, as well as Figure 4.

4. *Integration of accidents: The clarification in the manuscript text is noted. I recommend to reference the mentioned literature which attempts to estimate such impacts.*

Added text: *“Importantly, LCA inventories and assessment methods do not account for the risk of radiation exposure nuclear accidents⁴⁵. However, analysis by Hirschberg et al.⁴⁶ and others^{47,48} suggest that (a) in terms of lost life years, fatalities from accidents tend to be considerably smaller than health impacts from regular operation, and (b) that fatalities from nuclear accidents tend to be lower than those from fossil-based or hydro power.”* with references:

45. Hofstetter, P., Bare, J. C., Hammitt, J. K., Murphy, P. A. & Rice, G. E. Tools for Comparative Analysis of Alternatives: Competing or Complementary Perspectives? *Risk Analysis* **22**, 833–851 (2002).

46. Hirschberg, S. et al. Health effects of technologies for power generation: Contributions from normal operation, severe accidents and terrorist threat. *Reliability Engineering & System Safety* **145**, 373–387 (2016).

47. Volkart, K. et al. Interdisciplinary assessment of renewable, nuclear and fossil power generation with and without carbon capture and storage in view of the new Swiss energy policy. *International Journal of Greenhouse Gas Control* **54**, Part 1, 1–14 (2016).

48. Burgherr, P. & Hirschberg, S. Comparative risk assessment of severe accidents in the energy sector. *Energy Policy* **74**, S45–S56 (2014).

5. *I. 369: “the need for integrated global land management” is a rather unfortunate formulation. What is this supposed to look like in political reality? Perhaps replace by something along the lines of ‘the need for integrated land management across the globe’, or ‘international policy frameworks for sustainable land management’?*

Thanks for this suggestion. Our point is that in view of the indirect and cross-border effects international coordination is required. Reformulated to: *“This underlines the need for international policy frameworks for sustainable land management to overcome the tradeoff between climate change mitigation and conservation.”*

6. *I. 489: Part of the sentence is missing. ‘Corresponding to’ what?*

Resolved: *“In MAgPIE, residues available for bioenergy use are exogenously assumed to amount to 47 EJ in 2050, corresponding to around one third of total global bioenergy demand projected in the IAM climate change mitigation scenarios for 2050.”*

REVIEWERS' COMMENTS:

Reviewer #1 (Remarks to the Author):

The authors have satisfactorily addressed all my comments. No further changes or review is required.

Reviewer #2 (Remarks to the Author):

Almost all my comments have been addressed. However, the reply to my comment (5) does not seem to be reflected in the manuscript, as far as I can see (which is quite hard, as the authors have decided to omit line numbers in their reply). The "the need for international policy frameworks for sustainable land management" cannot be found anywhere.

We would like to thank the two Referees for their final comments and guidance throughout the review process. Our responses are in purple font.

Reviewer #1 (Remarks to the Author):

*The authors have satisfactorily addressed all my comments. No further changes or review is required.
Thank you!*

Reviewer #2 (Remarks to the Author):

Almost all my comments have been addressed. However, the reply to my comment (5) does not seem to be reflected in the manuscript, as far as I can see (which is quite hard, as the authors have decided to omit line numbers in their reply). The "the need for international policy frameworks for sustainable land management" cannot be found anywhere.

Our apologies for this glitch. The new text "This underlines the need for international policy frameworks for sustainable land management to overcome the tradeoff between climate change mitigation and conservation." is now included in the final version of the manuscript at lines 367-369 (clean version).